# Metabolic requirement for GOT2 in pancreatic cancer depends on environmental context

Samuel A Kerk[1], Lin Lin[2], Amy L Myers[2], Damien J Sutton[2], Anthony Andren[2], Peter Sajjakulnukit[1], Li Zhang[2], Yaqing Zhang[3], Jennifer A Jiménez[1], Barbara S Nelson[1,2], Brandon Chen[2], Anthony Robinson[4], Galloway Thurston[2], Samantha B Kemp[5], Nina G Steele[4], Megan T Hoffman[2], Hui-Ju Wen[2], Daniel Long[2], Sarah E Ackenhusen[6], Johanna Ramos[2], Xiaohua Gao[2], Zeribe C Nwosu[2], Stefanie Galban[7,8], Christopher J Halbrook[2], David B Lombard[9], David R Piwnica-Worms[10], Haoqiang Ying[11], Marina Pasca di Magliano[3,8], Howard C Crawford[2,8], Yatrik M Shah[2,8,12]*, Costas A Lyssiotis[1,2,8,12]*

[1]Doctoral Program in Cancer Biology, University of Michigan-Ann Arbor, Ann Arbor, United States; [2]Department of Molecular and Integrative Physiology, University of Michigan-Ann Arbor, Ann Arbor, United States; [3]Department of Surgery, University of Michigan-Ann Arbor, Ann Arbor, United States; [4]Department of Cell and Developmental Biology, University of Michigan-Ann Arbor, Ann Arbor, United States; [5]Molecular and Cellular Pathology Graduate Program, University of Michigan-Ann Arbor, Ann Arbor, United States; [6]Program in Chemical Biology, University of Michigan-Ann Arbor, Ann Arbor, United States; [7]Department of Radiology, University of Michigan, Ann Arbor, United States; [8]Rogel Cancer Center, University of Michigan, Ann Arbor, United States; [9]Department of Pathology and Institute of Gerontology, University of Michigan, Ann Arbor, United States; [10]Department of Cancer Systems Imaging, University of Texas MD Anderson Cancer Center, Houston, United States; [11]Department of Molecular and Cellular Oncology, University of Texas MD Anderson Cancer Center, Houston, United States; [12]Department of Internal Medicine, Division of Gastroenterology and Hepatology, University of Michigan, Ann Arbor, United States

*For correspondence:
shahy@umich.edu (YMS);
clyssiot@umich.edu (CAL)

**Competing interest:** The authors declare that no competing interests exist.

**ABSTRACT** Mitochondrial glutamate-oxaloacetate transaminase 2 (GOT2) is part of the malate-aspartate shuttle, a mechanism by which cells transfer reducing equivalents from the cytosol to the mitochondria. GOT2 is a key component of mutant KRAS (KRAS*)-mediated rewiring of glutamine metabolism in pancreatic ductal adenocarcinoma (PDA). Here, we demonstrate that the loss of GOT2 disturbs redox homeostasis and halts proliferation of PDA cells in vitro. GOT2 knockdown (KD) in PDA cell lines in vitro induced NADH accumulation, decreased Asp and α-ketoglutarate (αKG) production, stalled glycolysis, disrupted the TCA cycle, and impaired proliferation. Oxidizing NADH through chemical or genetic means resolved the redox imbalance induced by GOT2 KD, permitting sustained proliferation. Despite a strong in vitro inhibitory phenotype, loss of GOT2 had no effect on tumor growth in xenograft PDA or autochthonous mouse models. We show that cancer-associated fibroblasts (CAFs), a major component of the pancreatic tumor microenvironment (TME), release the redox active metabolite pyruvate, and culturing GOT2 KD cells in CAF conditioned media (CM) rescued proliferation in vitro. Furthermore, blocking pyruvate import or pyruvate-to-lactate reduction prevented rescue of GOT2 KD in vitro by exogenous pyruvate or CAF CM. However, these interventions failed to sensitize xenografts to GOT2 KD in vivo, demonstrating

the remarkable plasticity and differential metabolism deployed by PDA cells in vitro and in vivo. This emphasizes how the environmental context of distinct pre-clinical models impacts both cell-intrinsic metabolic rewiring and metabolic crosstalk with the TME.

## Editor's evaluation

This paper provides evidence that the environmental redox state shapes metabolic liabilities in pancreatic cancer cells. In vitro, pancreatic cancer cells rely on the malate-aspartate shuttle to regenerate oxidized electron acceptors in the cytosol that are required for cell proliferation; stromal cells within tumors secrete metabolites that serve as alternate electron acceptors, thereby reducing cancer cell reliance on the malate-aspartate shuttle.

## Introduction

Cancer cells depend on deregulated metabolic programs to meet energetic and biosynthetic demands (*Pavlova and Thompson, 2016*; *Vander Heiden and DeBerardinis, 2017*; *Martínez-Reyes and Chandel, 2021*). Metabolic therapies aim to preferentially target these dependencies (*Vander Heiden, 2011*). This approach has shown promise in preclinical models of pancreatic ductal adenocarcinoma (PDA) – one of the deadliest major cancers, notoriously resistant to anti-cancer therapies (*Ryan et al., 2014*; *Halbrook and Lyssiotis, 2017*). Pancreatic tumors are poorly vascularized and nutrient dysregulated (*Kamphorst et al., 2015*). Therefore, cancer cells commandeer metabolic pathways to scavenge and use nutrients (*Halbrook and Lyssiotis, 2017*; *Encarnación-Rosado and Kimmelman, 2021*). A wealth of recent literature has identified that this is mediated predominantly by mutant KRAS (KRAS*), the oncogenic driver in most pancreatic tumors (*Commisso et al., 2013*; *Ying et al., 2012*; *Son et al., 2013*; *Viale et al., 2014*; *Santana-Codina et al., 2018*; *Humpton et al., 2019*). The KRAS* has also been implicated in shaping the pancreatic tumor microenvironment (TME; *Tape et al., 2016*; *Kerk et al., 2021*). The PDA tumors exhibit a complex TME (*Storz and Crawford, 2020*; *Zhang et al., 2019*) with metabolic interactions between malignant, stromal, and immune cells enabling and facilitating tumor progression (*Lyssiotis and Kimmelman, 2017*). Recent successes in drug development have provided KRAS* selective inhibitors, and these are in various stages of preclinical and clinical testing. However, consistent with other targeted therapies, resistance inevitably occurs (*Janes et al., 2018*; *Jänne, 2022*). Therefore, disrupting downstream metabolic crosstalk mechanisms in PDA is a compelling combinatorial or alternative approach (*Cox et al., 2014*).

In support of this idea, previous work from our lab described that PDA cells are uniquely dependent on KRAS*-mediated rewiring of glutamine metabolism for protection against oxidative stress (*Son et al., 2013*). Mitochondrial glutamate-oxaloacetate transaminase 2 (GOT2) is implicated in this rewired metabolism in PDA. In normal physiology, GOT2 functions in the malate-aspartate shuttle (MAS), a mechanism by which cells transfer reducing equivalents between the cytosol and mitochondria to balance the two independent NADH pools and maintain redox balance (*Figure 1A*). The PDA cells driven by KRAS* divert metabolites from the MAS and increase flux through malic enzyme 1 (ME1) to produce NADPH (*Son et al., 2013*). Since this pathway is critical for PDA, we set out to evaluate GOT2 as a potential therapeutic target. Using metabolomics analyses and manipulation of the redox state in PDA cells, we discovered that loss of GOT2 in vitro induces intracellular NADH accumulation and reductive stress. These metabolic changes impair cellular growth, which can be rescued with chemical or genetic interventions that oxidize NADH. However, loss of GOT2 had no effect on tumor growth or initiation in immunocompromised or immunocompetent mouse models of PDA. Cancer cells use a complex cell-intrinsic rewiring and crosstalk with the TME to maintain redox homeostasis in vivo. These data emphasize an underappreciated role for GOT2 in pancreatic tumor redox homeostasis and illustrate the differential biochemical pathways and metabolic plasticity deployed by cancer cells in vivo.

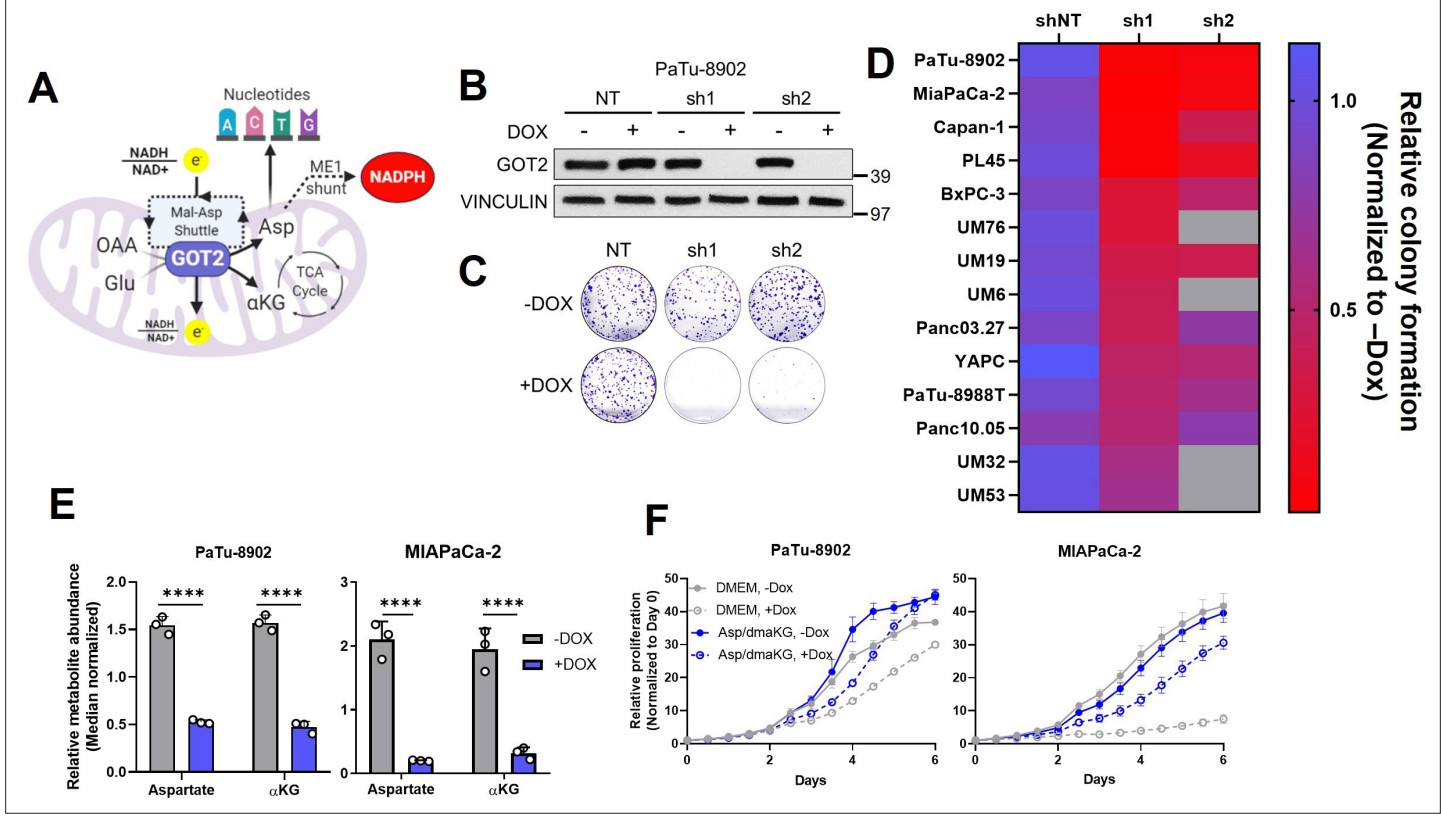

**Figure 1.** Glutamate-oxaloacetate transaminase 2 (GOT2) knockdown (KD) impairs in vitro pancreatic ductal adenocarcinoma (PDA) proliferation. (**A**) The metabolic roles of mitochondrial GOT2. OAA: oxaloacetate; Glu: glutamate; Mal: malate; Asp: aspartate; αKG: α-ketoglutarate; ME1: malic enzyme 1; TCA: tricarboxylic acid. (**B**) Immunoblot of GOT2 and VINCULIN loading control in PaTu-8902 cells after 1 µg/mL doxycycline (DOX) induction of two independent GOT2 (sh1, sh2) and non-targeting (NT) shRNAs for 3 days. (**C**) Representative images from colony formation assays in ishRNA PaTu-8902 cells −Dox (n=3) or +Dox (n=3). (**D**) Heatmap summarizing the relative colony formation ishRNA PDA cell lines −Dox (n=3) or +Dox (n=3), normalized to −Dox for each indicated shRNA. Representative images from colony formation assays and western blots presented in *Figure 1—figure supplement 1*. (**E**) Relative abundances of Asp and αKG in PaTu-8902 (left) and MIAPaCa-2 (right) ishGOT2.1 −Dox (n=3) or +Dox (n=3). (**F**) Relative proliferation of PaTu-8902 (left) and MIAPaCa-2 (right) ishGOT2.1 −Dox (n=3) or +Dox (n=3) cultured in normal media (Dulbecco's modified Eagle medium, DMEM) or supplemented with 20 mM Asp and 1 mM dimethyl-αKG (dmαKG). For all panels, data represent mean ± SD. *p<0.05, **p<0.01, ***p<0.001, and ****p<0.0001.

The online version of this article includes the following source data and figure supplement(s) for figure 1:

**Source data 1.** Full western blot images for *Figure 1B*.

**Figure supplement 1.** GOT2 knockdown (KD) impairs in vitro colony formation and proliferation of pancreatic cancer cell lines.

**Figure supplement 1—source data 1.** Full western blot images for *Figure 1—figure supplement 1B,E*.

**Figure supplement 2.** Metabolic effect of GOT2 KD in vitro in pancreatic cancer cell lines.

# Results

## Loss of GOT2 impairs PDA cell proliferation in vitro

To expand on our previous work studying GOT2 in PDA (*Son et al., 2013*) and to evaluate GOT2 as a potential therapeutic target, we generated a panel of PDA cell lines with doxycycline-inducible expression of either a control non-targeting shRNA (shNT) or two independent shRNAs (sh1, sh2) targeting the *GOT2* transcript. Cells cultured in media containing doxycycline (+Dox) exhibited a marked decrease in GOT2 protein expression compared to cells cultured in media without doxycycline (−Dox) (*Figure 1B*; *Figure 1—figure supplement 1A*). This knockdown (KD) was specific for GOT2, relative to the cytosolic aspartate aminotransaminase GOT1 (*Figure 1—figure supplement 1B*). Having validated GOT2 KD, we tested the importance of GOT2 for cellular proliferation. In general, GOT2 KD in PDA cells impaired colony formation (*Figure 1C and D*; *Figure 1—figure supplement 1C*) and proliferation (*Figure 1—figure supplement 1D*). Consistent with our previous

report (*Son et al., 2013*), GOT2 was not required for the proliferation of non-transformed pancreatic cell types (*Figure 1—figure supplement 1E,F*).

Since GOT2 has several vital metabolic roles in a cell (*Figure 1A*), the changes caused by decreased GOT2 expression in PDA cells were examined using liquid chromatography-coupled tandem mass spectroscopy (LC-MS/MS). Numerous changes in the intracellular metabolome of GOT2 KD cells were observed (*Figure 1—figure supplement 2A,B*). Of note, the products of the GOT2-catalyzed reaction, aspartate (Asp) and α-ketoglutarate (αKG), were decreased (*Figure 1E*), and supplementation of these metabolites rescued growth of GOT2 KD (*Figure 1F*). While these PDAC cell lines do not express Asp transporters, we confirmed that supraphysiological levels of Asp (20-fold excess) led to an increase in intracellular Asp (*Figure 1—figure supplement 2C*). In addition to reduced αKG, there was a disruption in tricarboxylic acid (TCA) cycle intermediates, consistent with a role for GOT2 in facilitating glutamine anaplerosis (*Figure 1—figure supplement 2D*).

## GOT2 KD perturbs redox homeostasis in PDA cells

Aside from the expected decrease in Asp and αKG, and the perturbation of the TCA cycle, closer examination of the GOT2 KD metabolomics dataset revealed an impairment in glycolysis with a node at glyceraldehyde 3-phosphate dehydrogenase (GAPDH), indicative of NADH reductive stress (*Figure 2A*). Examination of glycolytic rate via Seahorse Flux Analysis confirmed that glycolysis was indeed impaired in GOT2 KD cells (*Figure 2—figure supplement 1A*). The GAPDH reduces NAD+ to produce NADH, where a buildup of NADH product inhibits GAPDH activity. Indeed, metabolite pools in upstream glycolysis and branch pathways like the pentose phosphate pathway are increased, and those in downstream glycolysis are decreased (*Figure 2B*). In cultured PDA cells, the MAS transfers glycolytic reducing potential to drive the electron transport chain (ETC) and support the maintenance of cytosolic redox balance (*Figure 1A*). We thus hypothesized that GOT2 KD interrupted this shuttle, preventing the proper transfer of electron potential in the form of NADH between these two compartments. In support of this, GOT2 KD increased the intracellular ratio of NADH to NAD+ (*Figure 2C*).

NADH accumulation leads to reductive stress, which can be relieved if the cell has access to electron acceptors (*Gui et al., 2016*). Pyruvate is a notable metabolite in this regard, as it can accept electrons from NADH, producing lactate and regenerating NAD+, in a reaction catalyzed by lactate dehydrogenase (LDH). Therefore, we hypothesized that pyruvate could rescue the defect in cellular proliferation mediated by GOT2 KD. Indeed, culturing GOT2 KD cells in pyruvate rescued proliferation in a dose-dependent manner (*Figure 2D and E*; *Figure 2—figure supplement 1B*). Additionally, cells expressing a genetically encoded, fluorescent ATP sensor indicated that ATP levels dropped with GOT2 KD and were restored with pyruvate supplementation (*Figure 2—figure supplement 1C*), reflecting the link between TCA cycle activity, respiration, and oxidative phosphorylation. Furthermore, having identified a metabolite that permits in vitro proliferation of PDA cells without GOT2, we engineered CRISPR-Cas9 *GOT2* knockout (KO) cells for further investigation (*Figure 2—figure supplement 1D*). In support of the data generated using the doxycycline-inducible shRNA, *GOT2* KO impaired colony formation of PDA cells, which was similarly restored through extracellular pyruvate supplementation (*Figure 2—figure supplement 1E,F*).

The α-ketobutyrate (αKB) is another electron acceptor that turns over NADH in a mechanism analogous to pyruvate but without entering downstream metabolism in the same fashion as pyruvate (*Sullivan et al., 2015*). The αKB also rescued proliferation after GOT2 KD (*Figure 2F*; *Figure 2—figure supplement 1F*). This mechanism is dependent on NADH turnover and not NAD+ synthesis, as the NAD+ precursor nicotinamide mononucleotide (NMN) failed to rescue GOT2 KD (*Figure 2—figure supplement 1G*).

To test this further, GOT2 KD cells were engineered to express either doxycycline-inducible cytosolic or mitochondrial *Lactobacillus brevis* NADH oxidase (LbNOX), which uses molecular oxygen to oxidize NADH and produce water and NAD+ (*Figure 2—figure supplement 2A*; *Goodman et al., 2020*; *Titov et al., 2016*). Cytosolic LbNOX, but not mitochondrial LbNOX (mLbNOX), rescued proliferation of GOT2 KD cells (*Figure 2G*; *Figure 2—figure supplement 2B*). We confirmed the mLbNOX construct encoded a functional enzyme as defects in proliferation incurred by treatment with complex I inhibitor piericidin could be rescued with mLbNOX as reported previously (*Figure 2—figure supplement 2C*; *Titov et al., 2016*). LbNOX reversed the increased NADH/NAD+ ratio induced by GOT2 KD via an overall decrease in NADH levels (*Figure 2H*; *Figure 2—figure supplement 2D*). Furthermore,

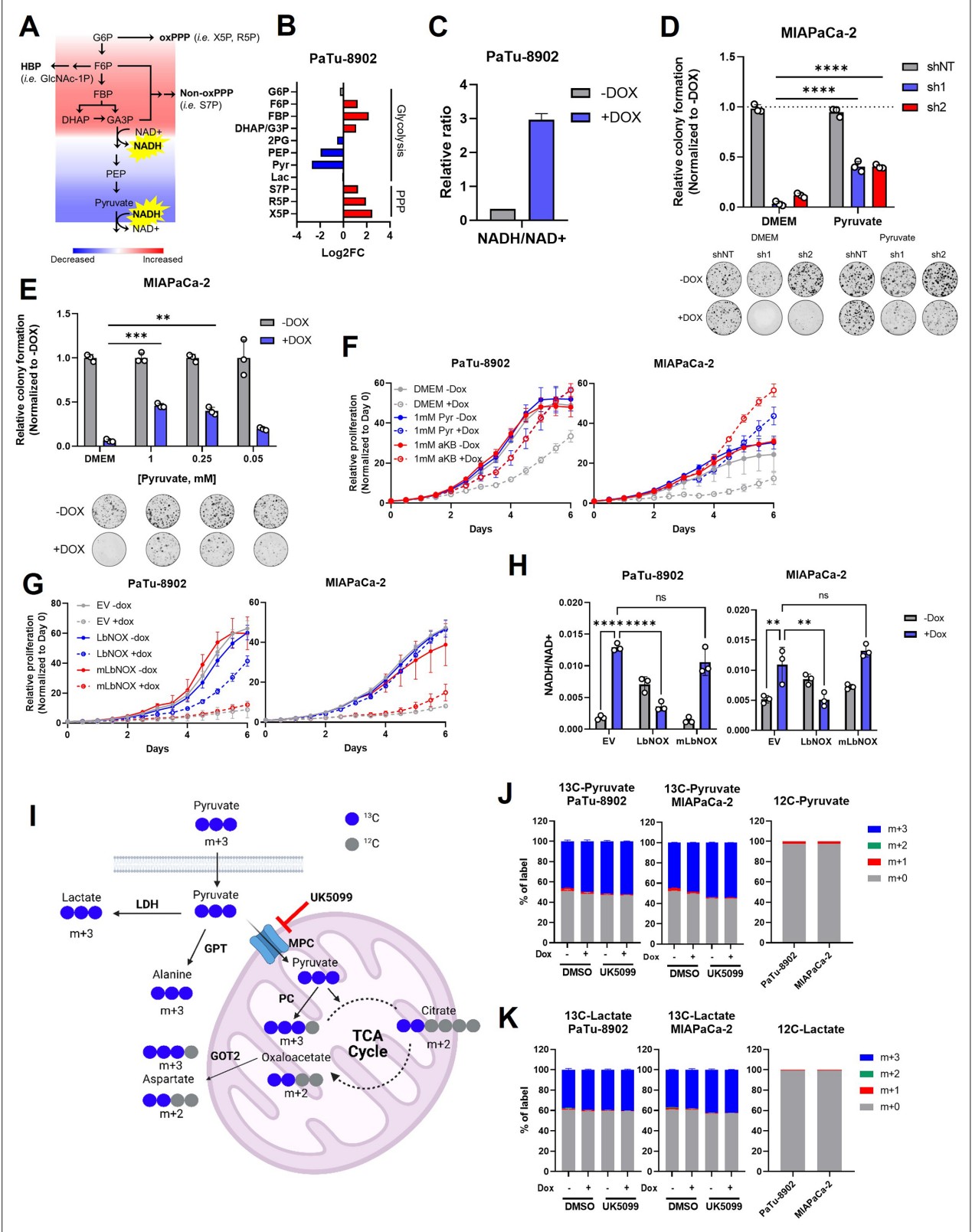

**Figure 2.** Glutamate-oxaloacetate transaminase 2 (GOT2) knockdown (KD) induces reductive stress, which can be ameliorated by NADH turnover. (**A**) Schematic of glycolytic signature induced by GOT2 KD-mediated NADH buildup and reductive stress. G6*P*: glucose-6-phosphate; F6*P*: fructose-6-phosphate; FBP: fructose-1,6-bisphosphate; DHAP: dihydroxyacetone phosphate; GA3*P*: glyceraldehyde-3-phosphate; PEP: phosphoenol pyruvate; oxPPP: oxidative pentose phosphate pathway; non-oxPPP: non-oxidative pentose phosphate pathway; X5*P*: xylulose-5-phosphate; R5*P*: ribose-5-

*Figure 2 continued on next page*

*Figure 2 continued*

phosphate; S7*P*: seduoheptulose-7-phosphate; HBP: hexosamine biosynthesis pathway; and GlcNAc-1*P*: N-acetylglucosamine 1-phosphate. (**B**) Relative fold changes in the indicated metabolites between PaTu-8902 ishGOT2.1 −Dox (n=3) and +Dox (n=3). 2PG: 2-phosphoglycerate; Pyr: pyruvate; and Lac: lactate. (**C**) Relative NADH/NAD+ ratio in PaTu-8902 ishGOT2.1 −Dox (n=3) and +Dox (n=3). (**D**) Relative colony formation of MIAPaCa-2 ishRNA −Dox (n=3) or +Dox (n=3) cultured in normal media (Dulbecco's modified Eagle medium, DMEM) or DMEM with 1 mM pyruvate, normalized to −Dox for each condition. (**E**) Relative colony formation of MIAPaCa-2 ishGOT2.1 −Dox (n=3) or +Dox (n=3) cultured in normal media (DMEM) or DMEM with the indicated concentrations of pyruvate (mM), normalized to −Dox for each condition. (**F**) Relative proliferation of PaTu-8902 (left) and MIAPaCa-2 (right) ishGOT2.1 −Dox (n=3) or +Dox (n=3) cultured in normal media (DMEM) or DMEM with 1 mM Pyr or α-ketobutyrate (αKB), normalized to Day 0 for each condition. (**G**) Relative proliferation of PaTu-8902 (left) and MIAPaCa-2 (right) ishGOT2.1 −Dox (n=3) or +Dox (n=3) expressing doxycycline-inducible empty vector (EV), cytosolic *Lactobacillus* NADH oxidase (LbNOX), or mitochondrial LbNOX (mLbNOX), normalized to Day 0 for each condition. (**H**) Relative NADH/NAD+ ratio of PaTu-8902 (left) and MIAPaCa-2 (right) ishGOT2.1 −Dox (n=3) or +Dox (n=3) expressing EV, LbNOX, or mLbNOX. (**I**) Schematic of 13C3-pyruvate into relevant metabolic pathways. 13C-carbon labels in blue, non-labeled carbon in gray. LDH: lactate dehydrogenase; GPT: glutamate-pyruvate transaminase; MPC: mitochondrial pyruvate carrier; and PC: pyruvate carboxylase. (**J–K**) Fractional labeling of intracellular pyruvate (J) or lactate (K) in PaTu-8902 and MIAPaCa-2 ishGOT2.1 −Dox (n=3) or +Dox (n=3) cultured in 1 mM 13C3-pyruvate and treated with DMSO vehicle control or 5 µM UK5099 (MPC inhibitor). Unlabeled controls presented at right. For all panels, data represent mean ± SD. *p<0.05, **p<0.01, ***p<0.001, and ****p<0.0001.

The online version of this article includes the following source data and figure supplement(s) for figure 2:

**Figure supplement 1.** GOT2 loss in vitro slows glycolysis and can be rescued by exogenous electron acceptors.

**Figure supplement 1—source data 1.** Full western blot images for *Figure 2—figure supplement 1D*.

**Figure supplement 2.** GOT2 KD can be rescued by cytosolic, but not mitochondrial, expression of LbNOX.

**Figure supplement 2—source data 1.** Full western blot images for *Figure 2—figure supplement 2A*.

**Figure supplement 3.** Cytosolic reduction of pyruvate to lactate is necessary for GOT2 KD.

the secreted pyruvate/lactate ratios in the media of LbNOX-expressing cells dramatically increased, indicating a resolution of the cytosolic NADH stress induced by GOT2 KD (*Figure 2—figure supplement 2E,F*). Moreover, the metabolic defects observed following GOT2 KD were ameliorated by cytosolic LbNOX activity, including an increase in Asp and αKG, normalization of TCA cycle metabolites, and the release of the glycolytic block at GAPDH (*Figure 2—figure supplement 2G*). The spatial control of the LbNOX system indicated that KD of mitochondrial GOT2 could be rescued by balancing the cytosolic NADH/NAD+ pool.

Next, we traced the metabolic fate of U13C-pyruvate in cells with GOT2 KD. Importantly, we also assessed the impact and metabolism of pyruvate in this system following inhibition of the mitochondrial pyruvate carrier (MPC) inhibitor UK5099, which blocks entry of pyruvate into the mitochondria. (*Figure 2I*). Media containing equimolar unlabeled glucose and U13C-pyruvate was used to prevent dilution of the pyruvate label by unlabeled pyruvate generated via glycolysis from the high glucose concentration used in normal media. This media formulation had no impact on the pyruvate GOT2 KD rescue phenotype (*Figure 2—figure supplement 3A*). In this experiment, roughly 50% of the intracellular pyruvate was labeled (*Figure 2J*), in line with the other half of the unlabeled pyruvate coming from glucose, and most of the labeled pyruvate was converted to lactate (*Figure 2K*). We observed modest labeling of citrate, Asp, and alanine from pyruvate (*Figure 2—figure supplement 3B-D*). While the labeled TCA cycle and branching pathway intermediates decreased dramatically with UK5099, this MPC inhibition had no effect on the pyruvate GOT2 KD rescue phenotype (*Figure 2—figure supplement 3E*).

Both lactate and alanine can fuel oxidation via the TCA cycle (*Hui et al., 2017*; *Sousa et al., 2016*), and Asp is critical for nucleotide production. Pyruvate can be converted into all three of these metabolites, yet supplementation with exogenous lactate, alanine, or nucleoside bases failed to rescue GOT2 KD (*Figure 2—figure supplement 3F-H*). These data collectively suggest a redox role for pyruvate-mediated rescue of GOT2 KD, as opposed to entering the mitochondria for ATP production or biosynthesis. These findings provide clear evidence using several orthogonal strategies that GOT2 KD results in accumulation of NADH pools and reductive stress in PDA cells.

## GOT2 is not required for PDA tumor growth in vivo

To test the effect of GOT2 KD on in vivo tumor growth, PDA cell lines were injected subcutaneously into the flanks of immunocompromised (non-obese diabetic (NOD) *scid* gamma; NSG) mice. We allowed the tumors to establish for 7 days, after which the mice were fed normal chow or doxycycline chow

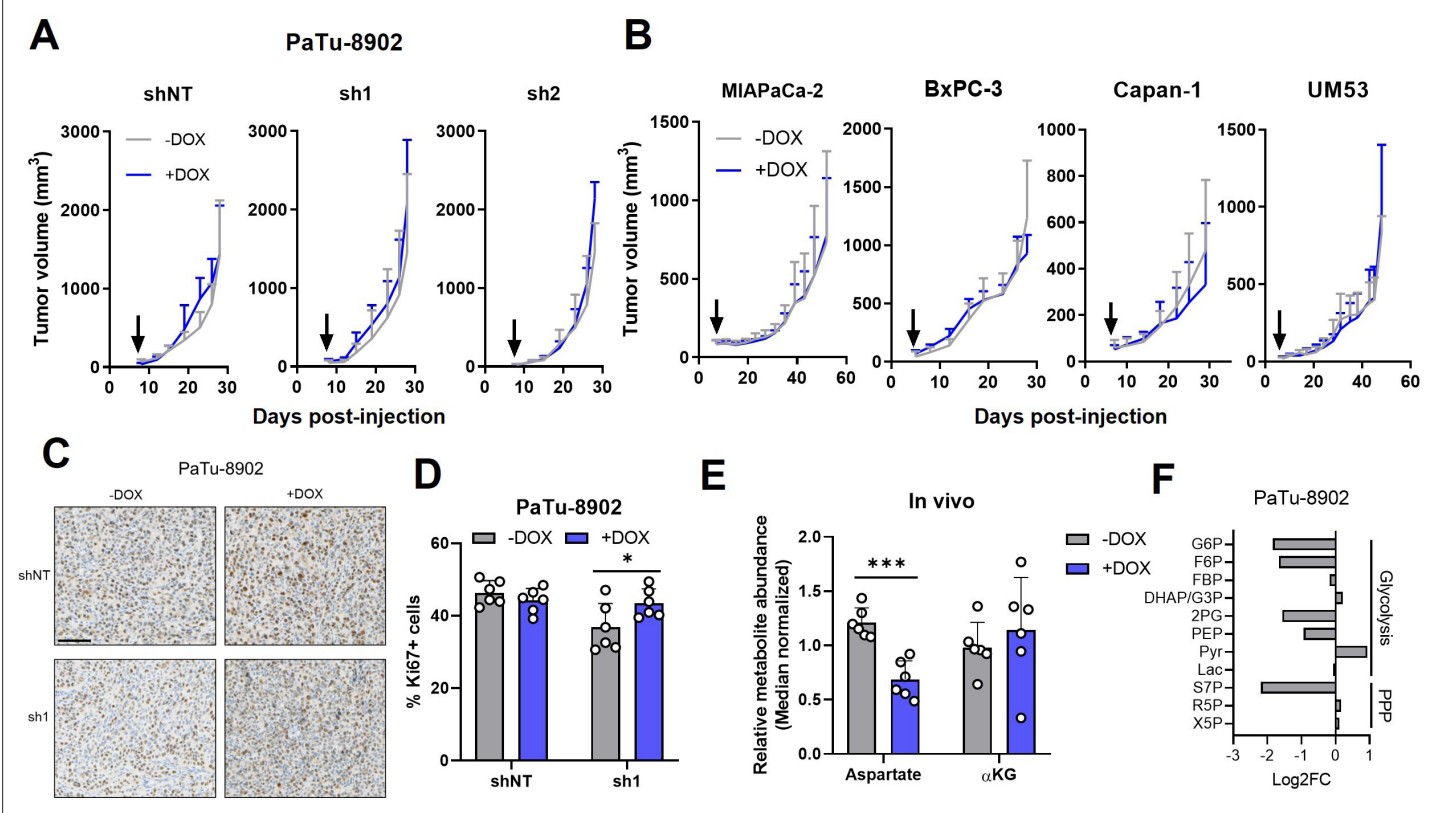

**Figure 3.** Glutamate-oxaloacetate transaminase 2 (GOT2) is not required for in vivo growth of pancreatic ductal adenocarcinoma (PDA) xenografts. (**A**) Tumor volumes of PaTu-8902 ishRNA flank xenografts in NOD scid gamma (NSG) mice fed normal chow (−Dox, n=6) or doxycycline chow (+Dox, n=6). Arrows indicate administration of Dox chow 1 week after PDA cell injection. (**B**) Tumor volumes of four additional PDA cell line ishGOT2.1 flank xenografts in NSG mice fed normal chow (−Dox, n=6) or doxycycline chow (+Dox, n=6). Arrows indicate administration of Dox chow 1 week after PDA cell injection. (**C**) Immunohistochemistry for Ki67 in flank xenograft tissue from PaTu-8902 ishRNA −Dox (n=6) or +Dox (n=6). Scale bar is 100 μm. (**D**) Quantification of Ki67+ cells in tissue depicted in (C). (**E**) Relative abundances of Asp and αKG in PaTu-8902 ishGOT2.1 −Dox (n=6) or +Dox (n=6) flank xenografts. (**F**) Relative fold changes in the indicated metabolites between PaTu-8902 ishGOT2.1 −Dox (n=6) and +Dox (n=6) flank xenografts. G6*P*: glucose-6-phosphate; F6*P*: fructose-6-phosphate; FBP: fructose-1,6-bisphosphate; DHAP: dihydroxyacetone phosphate; GA3*P*: glyceraldehyde-3-phosphate; 2PG: 2-phosphoglycerate; PEP: phosphoenol pyruvate; Pyr: pyruvate; Lac: Lactate; X5*P*: xylulose-5-phosphate; R5*P*: ribose-5-phosphate; S7*P*: seduoheptulose-7-phosphate; and *P*: pentose phosphate pathway. For all panels, data represent mean ± SD. *p<0.05, **p<0.01, ***p<0.001, and ****p<0.0001.

The online version of this article includes the following source data and figure supplement(s) for figure 3:

**Figure supplement 1.** GOT2 KD does not impede tumor growth or metabolism in vivo.

**Figure supplement 1—source data 1.** Full western blot images for *Figure 3—figure supplement 1B,G*.

(Dox chow) ad libitum. Surprisingly, despite the inhibitory in vitro phenotype, and robust suppression of GOT2 expression in vivo, PDA tumors from five different cell lines grew unimpeded with GOT2 KD (*Figure 3A and B*; *Figure 3—figure supplement 1A,B*). Nuclear Ki67 staining confirmed that tumors lacking GOT2 were proliferative and actually displayed a modest, but significant, increase in Ki67-positive nuclei (*Figure 3C and D*). To further examine the role of GOT2 in the proper tissue context, PDA cells were injected orthotopically into the pancreas of NSG mice, and tumors were allowed to establish for 7 days before feeding the mice regular or Dox chow. Similar to the flank model, GOT2 KD had no effect on the growth of orthotopic tumors (*Figure 3—figure supplement 1C*).

Having observed a discrepancy between in vitro and in vivo dependence on GOT2 for proliferation, the relative abundances of intracellular metabolites from flank tumors were analyzed via LC-MS/MS to compare the metabolic changes between cell lines and tumors following loss of GOT2. While GOT2 KD induced some changes in tumor metabolite levels, the affected metabolic pathways were distinct from those observed in vitro, bearing in mind we were comparing homogenous cell lines with heterocellular xenografts (*Figure 3E and F*; *Figure 3—figure supplement 1D-F*). Asp abundance was

significantly decreased, yet αKG levels remained constant (*Figure 3E*), and TCA cycle intermediates were unaffected (*Figure 3—figure supplement 1F*). This led us to initially hypothesize that PDA cells rewire their metabolism in vivo to maintain αKG levels when GOT2 is knocked down. However, upon examination of the expression of other αKG-producing enzymes in GOT2 KD tumors, we did not observe a compensatory increase in expression (*Figure 3—figure supplement 1G*). Certainly, expression does not always dictate metabolic flux, but these data led us to adopt an alternative, cell-extrinsic hypothesis to explain the different in vitro and in vivo GOT2 KD phenotypes. Finally, the glycolytic signature indicative of NADH stress was not observed in the metabolomics analysis from flank GOT2 KD tumors, further illustrating the differential dependence on GOT2 in PDA in vitro and in vivo (*Figure 3F*).

To evaluate the role of GOT2 in an immunocompetent model, we crossed the LSL-Kras^G12D;Ptf1a-Cre (KC) mouse with Got2^f/f mice to generate an LSL-Kras^G12D;Got2^f/f;Ptf1a-Cre mouse (KC-Got2) (*Figure 4A*, *Figure 4—figure supplement 1A*). Loss of *Got2* had no observable effect on the architecture of the healthy, non-transformed pancreas (*Figure 4—figure supplement 1B,C*). This is in support of our data demonstrating loss of GOT2 was not deleterious in human, non-malignant pancreatic cell types (*Figure 1—figure supplement 1F*).

KC-Got2 and KC controls were aged to 3, 6, and 12 months, at which point pancreata were harvested. We confirmed loss of GOT2 in the epithelial compartment via immunohistochemistry (IHC; *Figure 4B*). No differences were observed in the weights of pancreata between 3-month KC-Got2 and KC mice (*Figure 4C*). Scoring the hematoxylin and eosin (H&E) stained tissues from these groups by a blinded pathologist revealed that KC-Got2 mice had a significantly greater percentage of healthy acinar cells compared to KC controls (*Figure 4D and E*). However, no differences were observed in the percentages of acinar-ductal metaplasia or pancreatic intraepithelial grade between KC-Got2 and KC mice (*Figure 4E*). Additionally, we aged KC-Got2 mice to 6 months and compared the pancreata to matched 6-month KC historic controls, observing a slight decrease in weight for KC-Got2 pancreatic (*Figure 4F and G*). A histological analysis by a blinded pathologist did not identify a difference in number or severity of lesions (*Figure 4H*). In addition, both KC and KC-Got2 mice had progressed to carcinoma after aging for 1 year (*Figure 4I and J*). This suggests that loss of *Got2* does not affect the progression of PDA following transformation by oncogenic Kras.

## Cancer-associated fibroblast conditioned media supports colony formation in GOT2 KD cells in vitro

Human PDA tumors develop a complex microenvironment composed of a tumor-promoting immune compartment, a robust fibrotic response consisting of diverse stromal cell types, and a dense extracellular matrix (ECM; *Zhang et al., 2019*). While the flank tumor milieu in immunocompromised mice is less complex than that of a human PDA tumor, α-smooth muscle actin (αSMA) staining revealed that activated mouse fibroblasts comprised a substantial portion of the microenvironment in tumors regardless of GOT2 status (*Figure 5—figure supplement 1A*). Additionally, we and others have previously reported mechanisms by which cancer-associated fibroblasts (CAFs) in the stroma engage in cooperative metabolic crosstalk with pancreatic cancer cells (*Sousa et al., 2016*; *Zhao et al., 2016*; *Bertero et al., 2019*). So, we hypothesized that CAFs supported PDA metabolism following GOT2 KD. To investigate potential metabolic crosstalk in a simplified setting, PDA cells were cultured in vitro with CM from human CAFs (hCAFs). In support of our hypothesis, hCAF CM promoted colony formation in PDA cells with GOT2 KD in a dose-dependent manner (*Figure 5A and B*; *Figure 5—figure supplement 1B*). Furthermore, hCAF CM displayed a more pronounced colony formation rescue phenotype compared to CM from tumor-educated macrophages (TEMs) or from PDA cells (*Figure 5—figure supplement 1C*).

To begin to identify the factors in hCAF CM responsible for this effect, hCAF CM was boiled, filtered through a 3 kDa cut-off membrane, or subjected to cycles of freezing and thawing. In each of these conditions, hCAF CM supported colony formation in GOT2 KD cells, suggesting the relevant factor(s) was a metabolite (*Figure 5C*; *Figure 5—figure supplement 1D*). Therefore, the relative abundances of metabolites in hCAF CM were analyzed via LC-MS/MS (*Figure 5—figure supplement 1E*). Interestingly, when the metabolites were ranked in order of relative abundance, compared to the media control, pyruvate was one of the most differentially abundant metabolites released by hCAFs into the CM (*Figure 5D*). Since we used pyruvate-free Dulbecco's modified Eagle medium (DMEM)

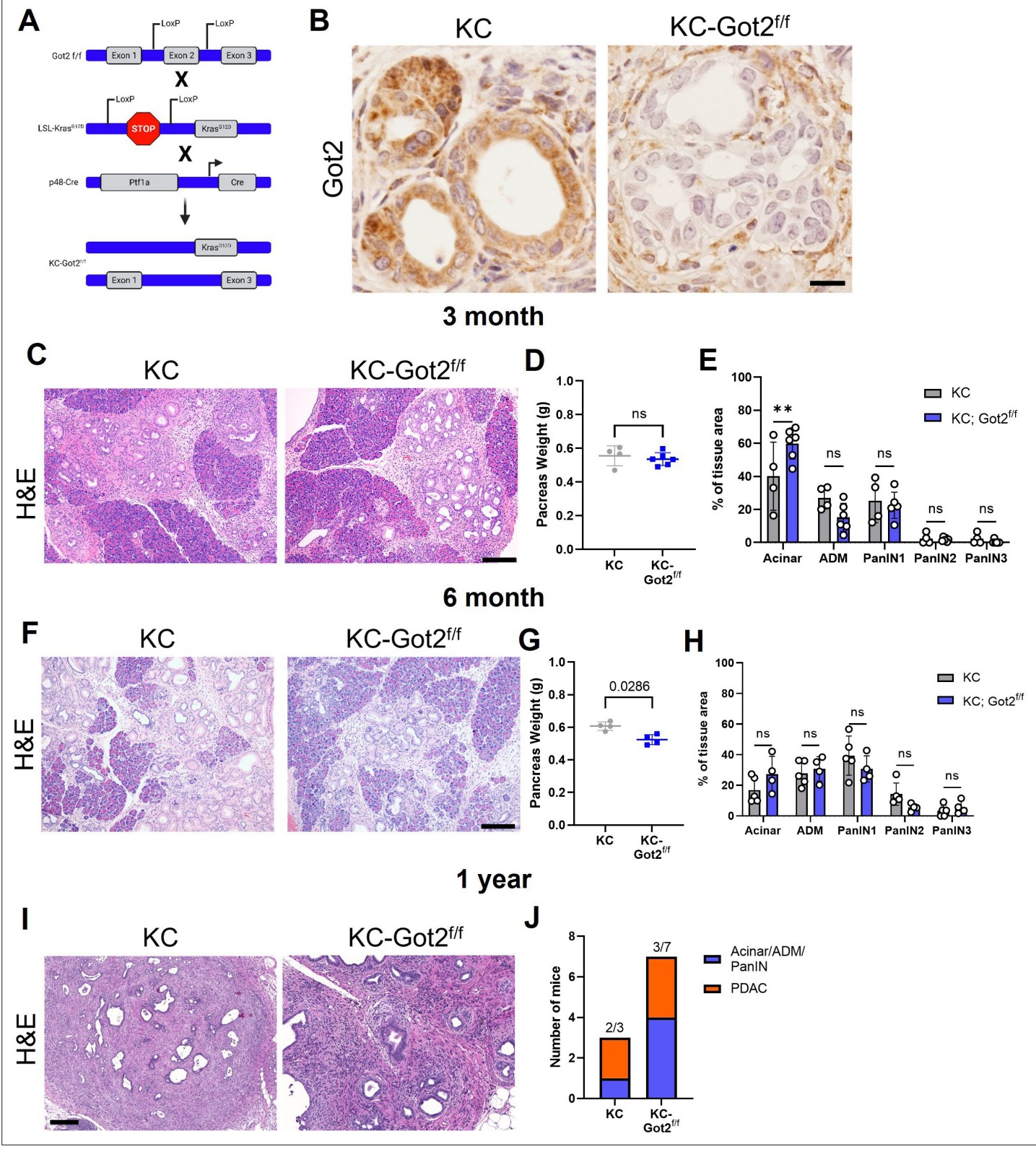

**Figure 4.** Glutamate-oxaloacetate transaminase 2 (GOT2) deletion does not impact on pancreatic ductal adenocarcinoma (PDA) tumorigenesis in an autochthonous model. (**A**) Got2 deletion (floxed exon 2) with expression of mutant Kras (LSL-Kras^G12D) driven by epithelial pancreas-specific Cre recombinase (p48-Cre) on an immunocompetent (C57B/6) background (KC-Got2). (**B**) Representative immunohistochemistry (IHC) for Got2 in pancreata from 3-month-old KC-Got2 or age-matched KC historic controls. Scale bar is 50 μM. (**C**) Representative hematoxylin and eosin (H&E) staining of

*Figure 4 continued on next page*

*Figure 4 continued*

pancreata from 3-month KC (n=4) or KC-Got2 (n=6) mice. Scale bar is 200 µM. (**D**) Pancreas weights of 3-month KC (n=4) or KC-Got2 (n=6) mice. (**E**) Quantitation of H&E staining from (C) of tissue area with healthy acinar cells, acinar-ductal metaplasia (ADM), or pancreatic intraepithelial (PanIN) lesion severity. (**F**) Representative H&E staining of pancreata from 6-month KC (n=5) or KC-Got2 (n=4) mice. Scale bar is 200 µM. (**G**) Pancreas weights of 6-month KC (n=4) or KC-Got2 (n=4) mice. (**H**) Quantitation of H&E staining from (F) of tissue area with healthy acinar cells, ADM, or PanIN lesion severity. (**I**) Representative H&E staining of pancreata from 1-year KC (n=3) or KC-Got2 (n=7) mice. Scale bar is 100 µM. (**J**) Number of KC or KC-Got2 tissue that had progressed to carcinoma at 1 year. For all panels, data represent mean ± SD. *p<0.05, **p<0.01, ***p<0.001, and ****p<0.0001.

The online version of this article includes the following figure supplement(s) for figure 4:

**Figure supplement 1.** Validation of Got2 loss in an autochthonous model of pancreatic tumorigenesis.

to culture hCAFs, to avoid overinterpreting this finding, we quantified the absolute concentration of pyruvate in hCAF CM at 250 µM, with some variability between batches (*Figure 5E*). This is a physiologically relevant concentration of pyruvate in serum collected from mice harboring pancreatic tumors (*Sullivan et al., 2019*), and 250 µM pyruvate rescued GOT2 KD in vitro (*Figure 2E*). Consistent with the idea of metabolite exchange, PDA cells cultured in hCAF CM had elevated levels of intracellular pyruvate (*Figure 5—figure supplement 1F*).

In the in vivo flank model, mouse fibroblasts infiltrate the pancreatic xenografts and engage in crosstalk with cancer cells. Therefore, we tested whether our findings with hCAFs were also applicable in mouse cancer-associated fibroblasts (mCAFs). The mCAFs isolated from a pancreatic flank xenograft in an NSG mouse also secreted pyruvate at similar levels to hCAFs in vitro (*Figure 5E*). Furthermore, mCAF CM promoted PDA colony formation following GOT2 KD (*Figure 5F*, *Figure 5—figure supplement 1G*). Since these mCAFs are the same CAFs encountered by PDA cells in our in vivo model, these data further support a mechanism by which CAFs compensate for loss of GOT2, by providing pyruvate to PDA cells lacking GOT2.

To better understand the production and release of pyruvate in CAFs, we traced glucose metabolism using uniformly carbon-labeled (U13C) glucose and LC-MS metabolomics. We demonstrate that the pyruvate released by CAFs was produced from glucose (*Figure 5G*), and, in support of previous studies (*Zhang et al., 2015*; *Lemons et al., 2010*), these CAFs displayed labeling patterns indicative of glycolytic metabolism (*Figure 5H*).

Aside from pyruvate, we also detected significantly elevated levels of Asp and αKG in CAF CM (*Figure 5—figure supplement 1E*). Since these are metabolites produced by GOT2, we next asked whether they were present at sufficient concentrations in CAF CM to compensate for loss of GOT2 in vitro. However, we quantified Asp at 15 µM and αKG at 50 µM in CAF CM (*Figure 5—figure supplement 2A*), well below the reported values from mouse serum (*Sullivan et al., 2019*). Furthermore, millimolar levels of Asp are required to impact intracellular levels, as PDA cell lines do not express an Asp transporter (*Birsoy et al., 2015*), and dimethyl-αKG must be used as αKG has limited membrane permeability (*Parker et al., 2021*). Nevertheless, we tested the rescue activity of reported serum levels of Asp and αKG (50 µM and 500 µM, respectively; *Sullivan et al., 2019*), again utilizing the dimethyl-αKG and compared this to serum levels of pyruvate (250 µM). Here we found that pyruvate rescued proliferation of GOT2 KD to a greater extent than the combination of Asp and αKG (*Figure 5—figure supplement 2B*). Therefore, we conclude that pyruvate, and not Asp or αKG, is responsible for the GOT2 KD rescue activity of CAF CM in vitro.

Like GOT2 inhibition, it is well established that inhibiting the activity of complex I of the ETC also results in an increase in NADH, which can be counteracted with extracellular pyruvate (*Gui et al., 2016*; *Sullivan et al., 2015*; *Titov et al., 2016*; *Birsoy et al., 2015*). Therefore, since pyruvate is highly abundant in hCAF CM, we hypothesized that PDA cells cultured in hCAF CM would be protected from complex I inhibitors. Indeed, both hCAF CM and extracellular pyruvate conferred resistance to PDA cells against the complex I inhibitors rotenone, phenformin, and IACS-010759 (*Figure 5—figure supplement 2C*; *Molina et al., 2018*). This points to a potential mechanism by which the TME could affect sensitivity to complex I inhibition, though more in depth studies are needed to test this finding further.

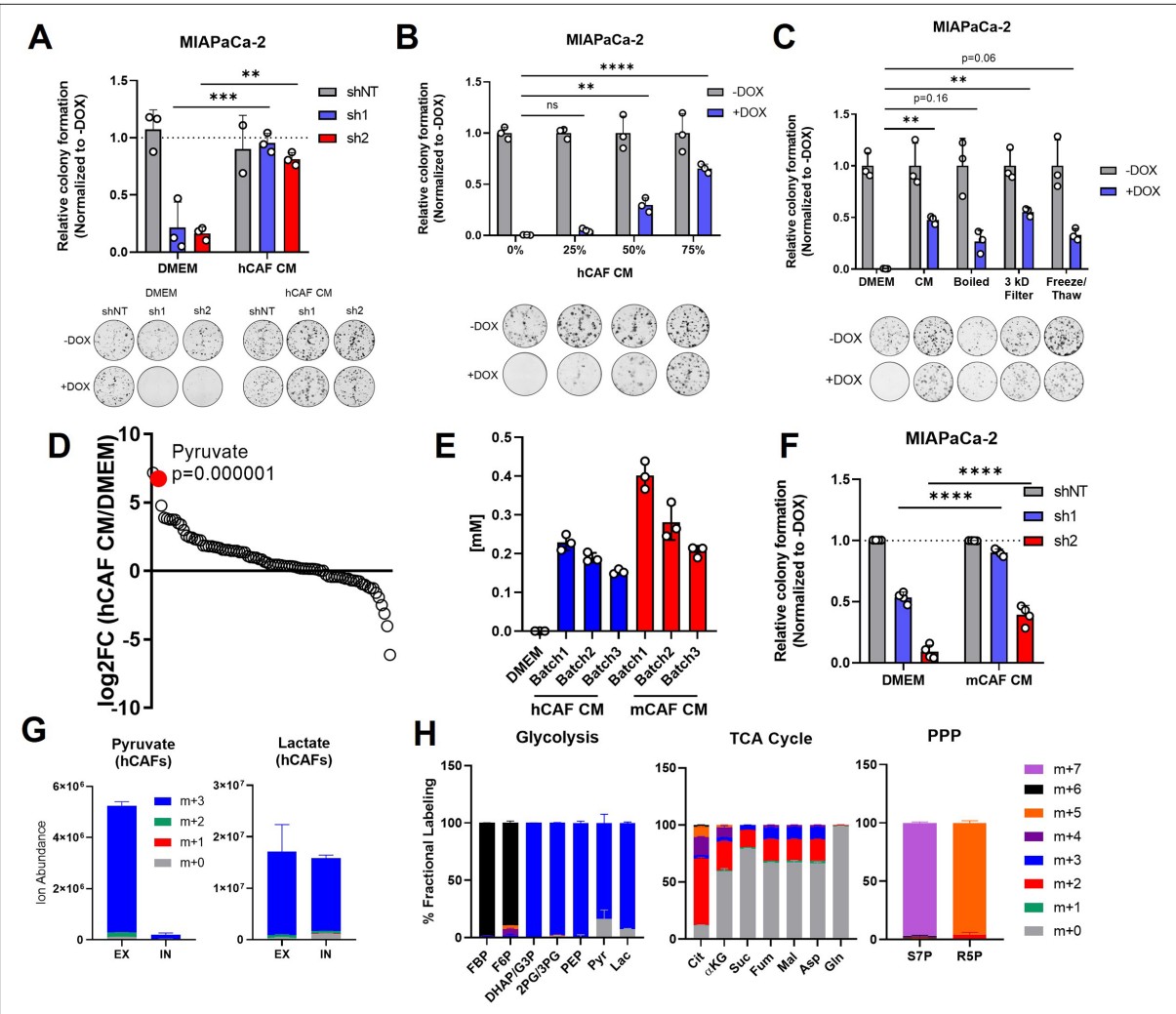

**Figure 5.** Pancreatic cancer-associated fibroblasts (CAFs) release pyruvate and compensate for loss of glutamate-oxaloacetate transaminase 2 (GOT2) in vitro. (**A**) Relative colony formation of MIAPaCa-2 ishGOT2.1 −Dox (n=3) or +Dox (n=3) cultured in normal media (Dulbecco's modified Eagle medium, DMEM) or human CAF (hCAF) conditioned media (CM) generated after 72 hr of conditioning, normalized to −Dox for each condition. (**B**) Relative colony formation of MIAPaCa-2 ishGOT2.1 −Dox (n=3) or +Dox (n=3) cultured in DMEM or indicated dilutions of hCAF CM, normalized to −Dox for each condition. (**C**) Relative colony formation of MIAPaCa-2 ishGOT2.1 −Dox (n=3) or +Dox (n=3) cultured in DMEM, mock-treated hCAF CM, boiled, 3 kDa filtered, or that subjected to freeze/thaw cycles. Data normalized to −Dox for each condition. (**D**) Ranked log2 fold changes of metabolite abundances in hCAF CM compared to pyruvate-free normal DMEM. (**E**) Absolute quantitation of pyruvate concentrations (mM) in three independently generated batches of hCAF or mouse CAF (mCAF) CM, including a pyruvate-free DMEM control. (**F**) Relative colony formation of MIAPaCa-2 ishGOT2.1 −Dox (n=3) or +Dox (n=3) cultured in normal media (DMEM) or mCAF CM, normalized to −Dox for each condition. (**G**) Ion abundance of extracellular (EX) or intracellular (IN) pyruvate (left) or lactate (right) isotopologues from hCAFs cultured with 25 mM 13C6-glucose. (**H**) Fractional labeling of intracellular glycolysis (left), tricarboxylic acid (TCA) cycle (middle), and pentose phosphate pathway (PPP, right) metabolites in hCAFs cultured with 25 mM 13C6-glucose. F6$P$: fructose-6-phosphate; FBP: fructose-1,6-bisphosphate; DHAP: dihydroxyacetone phosphate; G3$P$ : glyceraldehyde-3-phosphate; 2PG: 2-phosphoglycerate; 3PG: 3-phosphoglycerate; PEP: phosphoenol pyruvate; Cit: citrate; αKG: α-ketoglutarate; Suc: succinate; Fum: fumarate; Mal: malate; Gln: glutamine; Asp: aspartate; S7$P$: seduoheptulose-7-phosphate; and R5$P$: ribose-5-phosphate. For all panels, data represent mean ± SD. *p<0.05, **p<0.01, ***p<0.001, and ****p<0.0001.

The online version of this article includes the following figure supplement(s) for figure 5:

**Figure supplement 1.** CAF metabolism supports GOT2 KD pancreatic cancer cells in vitro.

**Figure supplement 2.** CAF-derived pyruvate confers resistance to complex I inhibitors in vitro.

## Inhibiting pyruvate uptake and metabolism blocks rescue of GOT2 KD in vitro

According to our model, PDA cells are more vulnerable to GOT2 KD or complex I inhibitors in a pyruvate-depleted environment or if pyruvate uptake were blocked. Pyruvate can be transported by four monocarboxylate transporter (MCT) isoforms (*Halestrap, 2013*; *Halestrap, 2012a*; *Halestrap and Wilson, 2012b*), and an analysis of the CCLE database suggests that PDA cell lines primarily express MCT1 and MCT4 (*Figure 6—figure supplement 1A*). Since MCT1 has a higher affinity for pyruvate than MCT4 (*Halestrap, 2012a*), we decided to focus on MCT1 as the transporter by which PDA cells import pyruvate (*Rao et al., 2020*; *Rao et al., 2021*). Indeed, PDA cells express significantly higher levels of MCT1, as compared to hCAFs (*Figure 6—figure supplement 1B*). Similarly, examining expression of MCT1 from a recently published single-cell analysis of a murine syngeneic orthotopic pancreatic tumor (*Steele et al., 2020*) indicated that PDA cells express high levels of MCT1 (*Figure 6—figure supplement 1C*).

Import/export of pyruvate and lactate through MCT1, as well as the intracellular redox state, is affected by extracellular concentrations of pyruvate and lactate. The absolute concentrations of pyruvate and lactate were measured in hCAF and mCAF CM to calculate the relative pyruvate/lactate ratio from in vitro GOT2 KD rescue experiments (*Figure 6A*). In parallel, the absolute levels of pyruvate and lactate, and the relative ratio, were measured in tumor interstitial fluid (TIF) from flank xenografts and in serum from host mice (*Figure 6B*). When the measured pyruvate and lactate levels from each of these permutations were added back to regular media, the pyruvate/lactate ratios mimicking CAF CM, serum, or TIF promoted the growth of GOT2 KD cells in vitro (*Figure 6C*). These data indicate the pyruvate/lactate ratios in CAF CM in vitro and in the in vivo TME are favorable for pyruvate import, possibly through MCT1.

Therefore, we hypothesized that blocking pyruvate import through MCT1 would render cells vulnerable to GOT2 KD. The small molecule AZD3965 has specificity for MCT1 over MCT4 (*Polański et al., 2014*; *Hong et al., 2016*; *Tasdogan et al., 2020*), therefore GOT2 KD cells were cultured in pyruvate or hCAF CM in the presence of AZD3965. In support of our hypothesis, MCT1 chemical inhibition reduced the pyruvate and hCAF CM rescue of GOT2 KD (*Figure 6D*).

For a genetic approach, MCT1 was knocked down in doxycycline-inducible GOT2 KD cells (*Figure 6—figure supplement 1D,F*). MCT1 KD modestly slowed the growth of GOT2 KD cells cultured in pyruvate or hCAF CM (*Figure 6—figure supplement 1E,G*). We reasoned that partial KD could explain why MCT1 KD had a more modest effect than chemical inhibition of MCT1. Therefore, we next generated MCT1 KO clones in GOT2 KD cells (*Figure 6E*). Indeed, MCT1 KO in GOT2 KD cells blocked pyruvate rescue (*Figure 6F*). We also demonstrate that MCT1 inhibition was most effective at physiological levels of pyruvate (250 μM; *Figure 6F*, *Figure 6—figure supplement 1H*). Furthermore, MCT1 KO cells cultured with U13C-pyruvate demonstrated reduced uptake of pyruvate, as measured by intracellular labeled pyruvate (*Figure 6G*). Since pyruvate is rapidly imported and converted to lactate, the reduced pyruvate uptake was observed more clearly in dramatically reduced levels of labeled lactate in MCT1 KO cells (*Figure 6G*). Lastly, MCT1 blockade was tested in combination with complex I inhibitors in PDA cells cultured in pyruvate or hCAF CM. The AZD3965 also reversed the rescue activity of pyruvate or hCAF CM in PDA cells treated with IACS-010759 (*Figure 6—figure supplement 1I*).

The reduction of pyruvate to lactate by LDH is the central mechanism in our model by which NAD+ is regenerated to support proliferation in GOT2 inhibited cells. Thus, we next asked whether inhibiting LDH activity could also prevent GOT2 KD rescue by pyruvate or hCAF CM. Our PDA cell lines highly expressed the LDHA isoform of LDH, as determined by western blotting (*Figure 6—figure supplement 1J*). As such, we utilized the LDHA-specific chemical inhibitor FX11 in this study (*Le et al., 2010*). In further support of our model, inhibiting LDHA with FX11 slowed the in vitro proliferation of GOT2 KD cells cultured in pyruvate or hCAF CM, relative to single agent controls (*Figure 6H*, *Figure 6—figure supplement 1K*).

Cumulatively, these data support an in vitro model whereby perturbation of mitochondrial metabolism with GOT2 KD or complex I inhibition disrupts redox balance in PDA cells. This can be restored through import of pyruvate from the extracellular environment and reduction to lactate to regenerate NAD+.

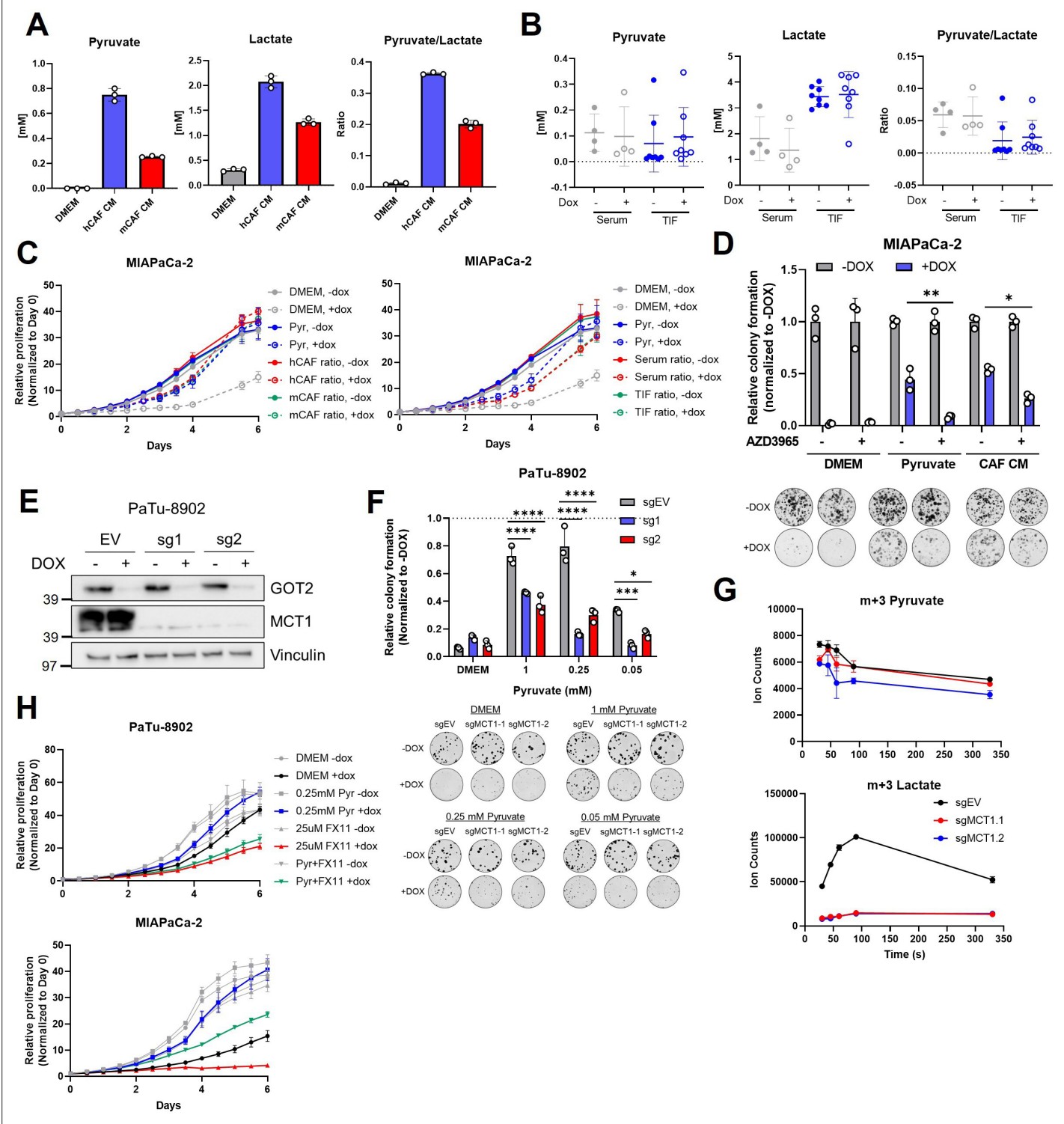

**Figure 6.** MCT1 inhibition prevents pyruvate-mediated restoration of redox balance in vitro after loss of glutamate-oxaloacetate transaminase 2 (GOT2). (**A**) Absolute quantitation of pyruvate (left) and lactate (middle), and the relative pyruvate/lactate ratio (right) in normal Dulbecco's modified Eagle medium (DMEM), human cancer-associated fibroblasts (hCAF) conditioned media (CM), and mouse CAF (mCAF) CM. (**B**) Absolute quantitation of pyruvate (left) and lactate (middle), and the relative pyruvate/lactate ratio (right) in serum or the tumor interstitial fluid (TIF) from NOD scid gamma (NSG) mice harboring PaTu-8902 ishGOT2.1 −Dox (n=8 tumors, 4 mice) or +Dox (n=8 tumors, 4 mice) flank xenografts. (**C**) Relative proliferation of MIAPaCa-2 ishGOT2.1 −Dox (n=3) or +Dox (n=3) cultured with the absolute levels and relative ratios of pyruvate/lactate in hCAF and mCAF CM (left) or serum/TIF (right) from (A) and (B), normalized to Day 0 for each condition. (**D**) Relative colony formation of MIAPaCa-2 ishGOT2.1 −Dox (n=3) or +Dox

*Figure 6 continued on next page*

*Figure 6 continued*

(n=3) cultured in normal media (DMEM), 1 mM pyruvate, or hCAF CM, and treated with DMSO control or 100 nM AZD3965, normalized to −Dox for each condition. (**E**) Immunoblots of GOT2, MCT1, and VINCULIN loading control in PaTu-8902 ishGOT2.1 expressing empty vector (EV), or two sgRNAs targeting MCT1 (sg1, sg2). (**F**) Relative colony formation of PaTu-8902 ishGOT2.1 −Dox (n=3) or +Dox (n=3) expressing EV, or two sgRNAs targeting MCT1 (sg1, sg2) and cultured in the indicated doses of pyruvate, normalized to −Dox for each condition. (**G**) Ion counts of intracellular m+3 pyruvate (top) and lactate (bottom) in PaTu-8902 expressing EV, or two sgRNAs targeting MCT1 (sg1, sg2) and cultured in 1 mM 13C3-pyruvate for the indicated time points. (**H**) Relative proliferation of PaTu-8902 (top) and MIAPaCa-2 (bottom) ishGOT2.1 −Dox (n=3) or +Dox (n=3) cultured in normal DMEM or 0.25 mM pyruvate and treated with DMSO vehicle control or 25 μM FX11, normalized to Day 0 for each condition. For all panels, data represent mean ± SD. *p<0.05, **p<0.01, ***p<0.001, and ****p<0.0001.

The online version of this article includes the following source data and figure supplement(s) for figure 6:

**Source data 1.** Full western blot images for *Figure 6E*.

**Figure supplement 1.** MCT1 and LDHA inhibition dampen pyruvate rescue of GOT2 KD in vitro.

**Figure supplement 1—source data 1.** Full western blot images for *Figure 6—figure supplement 1B,D,F,J*.

**Figure supplement 2.** MCT1 inhibition does not sensitize tumors to GOT2 KD in vivo.

**Figure supplement 2—source data 1.** Full western blot images for *Figure 6—figure supplement 2C,D,F,G,I*.

**Figure supplement 3.** LDHA inhibition does not sensitize tumors to GOT2 KD in vivo.

## Metabolic plasticity in vivo supports adaptation to combined GOT2 KD and inhibition of pyruvate metabolism

This proposed in vitro mechanism suggests that the unhampered growth of GOT2-inhibited tumors (*Figure 3A and B*) could be explained by uptake of pyruvate from the TME. In vitro, MCT1 KD had a modest effect on the proliferation of GOT2 KD cells cultured in pyruvate or hCAF CM. In contrast, we found that MCT1 KD had no effect on the growth of GOT2 KD subcutaneous xenografts (*Figure 6—figure supplement 2A,B*). A caveat of this finding was the observed detectable levels of MCT1 at endpoint (*Figure 6—figure supplement 2C,D*). However, MCT1 KO similarly did not impact the growth of GOT2 KD tumors in vivo, which is in stark contrast to the results observed in vitro whereby MCT1 KO blocked the rescue of colony formation in GOT2 KD cells by pyruvate (*Figure 6—figure supplement 2E*). In these tumors, MCT1 was not detectable at endpoint and we did not observe a compensatory increase in MCT4 expression in either cell lines or xenografts with MCT1 KO (*Figure 6—figure supplement 2F,G*). Similarly, administration of AZD3965 did not impact the growth of GOT2 KD xenografts, again contrasting our in vitro data showing strong blockade of GOT2 KD rescue by pyruvate or hCAF CM with MCT1 chemical inhibition (*Figure 6—figure supplement 2E-G*). Lastly, unlike in vitro, FX11 had no effect on GOT2 KD growth in vivo (*Figure 6—figure supplement 3A,B*).

## Discussion

GOT2 is an essential component of the MAS (*Borst, 2020*), which we now demonstrate is required for redox homeostasis in PDA cells in vitro. KD of GOT2 in vitro disrupts the MAS and renders PDA cells incapable of transferring reducing equivalents between the cytosol and mitochondria, leading to a cytosolic accumulation of NADH. This predominantly impacts the rate of glycolysis, an NAD+-coupled pathway, with secondary impacts on mitochondrial metabolism that together slow the proliferation of PDA cells in vitro. For instance, GOT2 feeds into the ME1 shunt, which we demonstrated previously also produces pyruvate and sustains intracellular NADPH levels (*Son et al., 2013*). Extracellular supplementation of electron acceptors like pyruvate and αKB, or the expression of a cytosolic NADH oxidase, relieves this NADH reductive stress and the associated pathway feedback inhibition.

In striking contrast to the in vitro data presented here, we also illustrate that GOT2 KD does not affect the growth of PDA tumors in vivo, potentially because electron acceptors in the TME can restore redox homeostasis. Indeed, pyruvate is present in mouse serum at 250 μM (*Sullivan et al., 2019*), a concentration which is sufficient to compensate for GOT2 KD in vitro. Furthermore, we also demonstrate that pancreatic CAFs release pyruvate in vitro, which can be utilized by PDA cells. This is supported by previous findings in CAFs from other cancers (*Sakamoto et al., 2019*). Therefore, a source of pyruvate is available to PDA tumors, either from circulation or potentially from the CAFs.

This led us to hypothesize that blocking pyruvate uptake and metabolism would deprive PDA cells of a critical means to relieve the NADH stress mediated by GOT2 KD. In vitro, this hypothesis was

supported by data illustrating that either inhibition of MCT1 or LDHA blocked the GOT2 KD rescue activity of pyruvate and CAF CM. However, this proved to be more complicated in vivo, as neither approach successfully sensitized tumors to GOT2 KD. We believe this could be explained by several mechanisms. First, previous work in KRAS*-driven non-small cell lung cancer reported differential dependencies on glutaminolysis in vitro in cell lines versus in vivo in lung tumors. Glutaminase inhibition was ineffective in vivo in these models as lung tumors primarily utilized glucose as a carbon source for the TCA cycle instead of glutamine. If this mechanism is active in our models, glucose, and not glutamine catabolism through GOT2, would fuel the TCA cycle. Second, while MCT4 has a lower affinity for pyruvate than MCT1, it can transport pyruvate (*Halestrap, 2012a*). It is also highly expressed in the cell lines used here (*Figure 6—figure supplement 2F,G*) and has been shown to confer resistance to cancer cells against MCT1 inhibition (*Bonglack et al., 2021*; *Hong et al., 2016*). Thus, dual inhibition of MCT1 and MCT4 may be required to effectively block pyruvate uptake. Third, even with sufficient MCT blockade, PDA cells could still obtain pyruvate through other processes, such as macropinocytosis (*Kamphorst et al., 2015*; *Commisso et al., 2013*). Fourth, while our study focuses on pyruvate, numerous circulating metabolites can function as electron acceptors and could relieve the intracellular accumulation of NADH if cancer cells are unable to import pyruvate (*Hui et al., 2020*). Fifth, reduction of pyruvate to lactate by LDH is not the only reaction by which NAD+ is regenerated. Recent studies have identified how serine pathway modulation, polyunsaturated fatty acid synthesis, and the glycerol-phosphate shunt all contribute to NADH turnover (*Yang et al., 2020*; *Kim et al., 2019*; *Liu et al., 2021*). Future work remains to assess these mechanisms in PDA cells in vivo and if they compensate for an impaired MAS. Nevertheless, our data emphasize that redox homeostasis is a vital aspect of cancer cell metabolism and is maintained through a complex web of intracellular compensatory pathways and extracellular interactions.

Aside from its broader role in redox balance, GOT2 is also a prominent source of Asp in PDA cells, and we demonstrate that GOT2 inhibition dramatically decreases Asp levels both in vitro and in vivo. Previous studies have shown that Asp availability is rate limiting in rapidly proliferating cells (*Sullivan et al., 2015*; *Birsoy et al., 2015*; *Sullivan et al., 2018*; *Garcia-Bermudez et al., 2018*). In our models, in contrast to physiological pyruvate concentrations simultaneous treatment with supraphysiological doses of both Asp and membrane-permeable dimethyl-αKG were required to provide a rescue of PDA cell proliferation in the absence of GOT2. Additionally, free Asp is available at low micromolar concentrations and has limited uptake capacity of PDA cells (*Sullivan et al., 2019*). Supplementation of physiologically relevant concentrations of Asp and αKG afforded a modest rescue compared to physiological pyruvate. Furthermore, exogenous protein supplementation with bovine serum albumin, another potential source for Asp, failed to rescue GOT2 KD in vitro (*Figure 6—figure supplement 3C*). While non-specific engulfment of ECM via macropinocytosis in vivo could supply PDA cells with Asp (*Garcia-Bermudez et al., 2022*), meeting a vital requirement for biosynthesis, we propose this does not address overall redox imbalance. Pyruvate, on the other hand, regenerates NAD+ allowing broader metabolic processes to resume, including Asp production and nucleotide biosynthesis. In support of this, recent work in myoblasts demonstrated that while complex I inhibition with piericidin increased the NADH/NAD+ ratio leading to depletion of Asp, adding Asp back to the system neither restored redox balance nor induced proliferation (*Mick et al., 2020*).

Our work also highlights the emerging metabolic role of CAFs in PDA. Recent studies have shown that CAFs engage in cooperative metabolic crosstalk with cancer cells in many different tumor types (*Lyssiotis and Kimmelman, 2017*; *Sousa et al., 2016*; *Zhao et al., 2016*; *Sanford-Crane et al., 2019*; *Schwörer et al., 2019*). We add to this body of literature by demonstrating that CAFs release pyruvate, which is taken up and utilized by PDA cells. However, much remains to be discovered about CAF metabolism how they contribute to redox homeostasis in tumors. Some types of activated fibroblasts are known to be highly glycolytic (*Zhang et al., 2015*; *Lemons et al., 2010*), an observation supported by our data. Yet the advent of single-cell RNA sequencing in murine and human pancreatic tumor models has led to a recent appreciation for the heterogeneity of CAFs (*Elyada et al., 2019*; *Ohlund et al., 2017*; *Neuzillet et al., 2019*; *Hosein et al., 2019*; *Helms et al., 2022*; *Hutton et al., 2021*). The newly identified iCAF, myCAF, and apCAF populations have distinct functions in a pancreatic tumor (*Elyada et al., 2019*) and likely employ distinct metabolism to carry out these functions. Much more remains to be uncovered regarding competitive or cooperative interactions between PDA cells and the various CAF subpopulations, including which subtype(s) are responsible for pyruvate release.

Lastly, this work suggests that the role of the TME should be considered when targeting cancer metabolism. Indeed, approaches like disrupting the MAS shuttle via GOT2 KD or blocking complex I with small molecule inhibitors were less effective in vitro when PDA cells were cultured in supplemental pyruvate or CAF CM. These data are relevant since numerous mitochondrial inhibitors are currently in clinical trials against solid tumor types (NCT03291938, NCT03026517, NCT03699319, and NCT02071862). Previous studies have also shown that complex I inhibitors are more effective in combination with AZD3965 (*Beloueche-Babari et al., 2017*), a selective inhibitor of MCT1, though our work and others indicate that the status of other MCT isoforms should also be considered. Furthermore, the abundance of CAFs present in a tumor, as well as the level of circulating pyruvate in the patient, could predict outcomes for treatment with metabolic therapies that lead to redox imbalance. Targeting pancreatic cancer metabolism is an alluring approach, and a more detailed understanding of the metabolic crosstalk occurring in a pancreatic tumor can shed light on potential resistance mechanisms and inform more effective metabolic therapies.

## Materials and methods
### Cell culture

MIAPaCa-2, BxPC-3, Capan-1, Panc03.27, Panc10.05, PL45, and HPNE cell lines were obtained from ATCC. PaTu-8902, PaTu-8988T, and YAPC cells lines were obtained from DSMZ. UM6, UM19, UM28, UM32, UM53, and UM76 were generated from primary patient tumors at the University of Michigan. Human pancreatic stellate cells (also described here as hCAFs) were a generous gift from Rosa Hwang (*Hwang et al., 2008*). The mCAFs were isolated as described below. All cell lines were cultured in high-glucose DMEM (Gibco) without pyruvate and supplemented with 10% fetal bovine serum (Corning). A 0.25% trypsin (Gibco) was used to detach and passage cells. Cell lines were tested regularly for mycoplasma contamination using MycoAlert (Lonza). All cell lines in this study were validated for authentication using STR profiling via the University of Michigan Advanced Genomics Core. L-Aspartic acid (Sigma), dimethyl-α-kG (Sigma), adenine (Sigma), guanine (Sigma), thymine (Sigma), cytosine (Sigma), sodium pyruvate (Invitrogen), αKB (Sigma), NMN (Sigma), L-alanine (Sigma), and sodium lactate (Sigma) were used at the indicated concentrations. UK5099, AZD3965, and phenformin were purchased from Cayman chemical, Rotenone from Sigma, FX11 from MedChem Express, and IACS-010759 was generously provided by Dr Giulio Draetta, University of Texas MD Anderson Cancer Center.

### Doxy-inducible shGOT2 cells

The tet-pLKO-puro plasmid was obtained from Dmitri Wiederschain via Addgene (#21915). Oligonucleotides encoding sense and antisense shRNAs (shGOT2.1-TRCN0000034824, shGOT2.2-TRCN0000034825) targeting *GOT2* (NM_002080.4) were synthesized (Integrated DNA Technologies), annealed, and cloned at AgeI and EcoRI sites according to the Wiederschain protocol (*Wiederschain et al., 2009*). A tet-pLKO non-targeting control vector (shNT-ccggcaacaagatgaagagcaccaactcga gttggtgctcttcatcttgttgttttt) was constructed using the same strategy. The tet-pLKO-shGOT2 and tet-pLKO-shNT lentiviruses were produced by the University of Michigan Vector Core using purified plasmid DNA. Stable cell lines were generated through transduction with optimized viral titers and selection with 1.5 µg/mL puromycin for 7 days.

### GOT2 or MCT1 KO cells

*GOT2* or *SLC16A1* (MCT1) KO PDA cell lines were generated using a CRISPR-Cas9 method described previously (*Ran et al., 2013*). Briefly, sgRNA oligonucleotide pairs were obtained from the Human GeCKO Library (v2, 3/9/2015). For *GOT2* KO (sg1 (Fwd), 5'-CACCgAAGCTCACCTTGCGGACGCT-3' , (Rev) 5'-AAACAGCGTCCGCAAGGTGAGCTTc; sg2 (Fwd) 5'-CACCgCGTTCTGCCTAGCGTCCGCA -3', (Rev) 5'-AAACTGCGGACGCTAGGCAGAACGc-3'), and for MCT1 KO (sg1 (Fwd) 5'- CACCgTGG GCCCGATTGGTCGCATG-3', (Rev) 5'- AAACCATGCGACCAATCGGGCCCAc; sg2 (Fwd) 5'- CACCg TTTCTACAAGAGGCGACCAT-3', (Rev) 5'- AAACATGGTCGCCTCTTGTAGAAAc-3') were cloned into the pSpCas9(BB)–2A-Puro plasmid (PX459, v2.0; Addgene, #62988), transfected in PDA cell lines, and selected in puromycin for 7 days. Cells were then seeded into 96 well plates at a density of 1 cell per

well, and individual clones were expanded. *GOT2* KO cells were maintained in 1 mM pyruvate for this entire process. The *GOT2* KO was verified via western blot. Cells transfected with the empty PX459 vector were used as controls.

## MCT1 KD cells

Cells were transduced with 8 µg/mL polybrene and lentivirus containing the pGFP-C-shLenti plasmid (Origine, #TR30023) containing an shCTR sequence, shMCT1.1 (TL309405A [5′-GAGGAAGAGACCA GTATAGATGTTGCTGG-3′]), or shMCT1.2 (TL309405B [5′-ATCCAGCTCTGACCATGATTGGCAA GTAT-3′]). These plasmids were a generous gift from Dr Sean Morrison (*Tasdogan et al., 2020*). The cells were then centrifuged at 1000× g for 60 min at room temperature. Transduced cells were then expanded and sorted on the MoFlo Astrios (Beckman-Coulter). The GFP+ cells were collected and expanded before verification of MCT1 KD via western blotting.

## Transduction of LbNOX/mLbNOX

pINDUCER (Addgene, #44014) plasmids containing GFP, LbNOX, or mitoLbNOX were obtained from Dr Haoqing Ying, MD Anderson. Plasmids were sequenced and transfected along with lentiviral packaging plasmids into HEK293FT cells with Lipofectamine 3000 (Thermo Fisher) per manufacturer's instructions. Virus was collected after 48 hr and filtered through a 0.2 µm filter. PaTu-8902 and MIAPaCa-2 iDox-shGOT2.1 cells were seeded in 6 well plates at 250,000 cells/well, transduced with the indicated vectors, and selected in G418 at 500 µg/mL for 7 days. Expression of Flag-tagged LbNOX or mitoLbNOX was confirmed by western blot with a Flag antibody after culturing cells in 1 µg/mL doxycycline for 3 days.

## Luciferase-expressing cells

MIAPaCa-2 iDox-shGOT2.1 cells were transduced with the FUGW-FL (EF1a-luc-UBC6-EGFP) lentiviral vector constructed previously (*Smith et al., 2004*), and GFP+ cells were selected via flow cytometry. Luciferase activity was confirmed following transduction and selection with an in vitro luciferase assay and detection on a SpectraMax M3 microplate reader (Molecular Devices).

## ATP fluorescent sensor/Incucyte growth assays

PaTu-8902 and MIAPaCa-2 iDox-GOT2.1 cells were transduced with CytoATP or CytoATP non-binding control vectors using the CytoATP Lentivirus Reagent Kit (Sartorious, #4772) and polybrene transfection reagent (Thermo Fisher) and selected for 7 days in 2 µg/mL puromycin. For proliferation and rescue experiments, cells were incubated in an Incucyte (Sartorious) equipped with a Metabolism Optical Module, where the ratio of ATP binding was detected and normalized to the non-binding control cells. Proliferation rate was determined by the percent confluence detected in the phase channel of the Incucyte normalized to Day 0 for each condition.

## Isolating mouse CAFs

UM2 subcutaneous xenografts from NSG mice were isolated and prepared in the laboratory of Dr Diane Simeone, as reported previously (*Li et al., 2007*), and single-cell suspensions were plated and cultured in vitro. Mouse CAFs were separated from human pancreatic UM2 cancer cells using the Mouse Cell Depletion Kit (MACS Miltenyi Biotec) according to the manufacturer's instructions.

## Conditioned media

CM was generated by splitting cells at ~90% confluence in a 10 $cm^2$ plate into four 15 $cm^2$ plates containing a final volume of 27 mL of growth media and incubating for 72 hr at 37°C, 5% $CO_2$. Afterward, the media was collected in 50 mL conical tubes, centrifuged at 1000 rpm for 5 min to remove any detached cells or debris, and divided into fresh 15 mL conical tubes in 10 mL aliquots for long-term storage at –80°C. For all CM experiments, unless indicated otherwise, growth media was mixed with CM for a final ratio of 75% CM to 25% fresh growth media.

For the experiments in *Figure 5C* and *Figure 5—figure supplement 1D*, CM was manipulated as follows. For boiling, the CM tubes were placed in a water bath at 100°C for 10 min. To filter out factors

>3 kDa, the CM were transferred to a 3 kDa filter (Millipore) and centrifuged at 15,000 rpm in 30 min increments until all the CM had passed through the filter. To expose the CM to freeze-thaw cycles, the tubes containing the CM were thawed for 30 min in a 60°C water bath, and then frozen at –80°C for 30 min. This was repeated two more times for a total of three freeze-thaw cycles.

## Colony formation assays

Cells were seeded in 6 well plates at 200–400 cells per well in 2 mL of growth media and incubated overnight at 37°C, 5% $CO_2$. The next day, the growth media was aspirated and fresh media containing the indicated compounds was added to the cells. Doxycycline was used at 1 µg/mL for all assays. For each assay, cells were incubated in the indicated conditions for 10 days, with the media and doxycycline changed every 3 days. After 10 days, the media was aspirated, the wells were washed once with PBS, and the cells were fixed in 100% methanol for 10 min. Next, the methanol was removed, and the cells were stained with 0.4% crystal violet for 10 min. Finally, the crystal violet was removed, the plates were washed under running water and dried on the benchtop overnight. The next day, images were taken of the plates with a Chemidoc BioRad imager and quantified using the ColonyArea plugin in ImageJ, as described previously (*Guzmán et al., 2014*).

## Proliferation assays

Cells were pre-treated for 3 days with 1 µg/mL doxycycline before seeding in 96 well plates at 1000 cells/well in 80 µL of media and incubated overnight at 37°C, 5% $CO_2$. The next day, 150 µL of the indicated treatment media was added to the appropriate wells, and the cells were incubated for 6–7 more days, with a media change on day 3. Cell proliferation was determined by live cell imaging for the duration of the assay, or using CyQUANT (Invitrogen) at endpoint according to the manufacturer's instructions, and detecting fluorescence on a SpectraMax M3 microplate reader (Molecular Devices).

## Glycolytic rate assay

PaTu-8902 iDox-shGOT2.1 cells that had been cultured in 1 µg/mL doxycycline for 3 days were seeded at $2 \times 10^4$ cells/well in 80 µL/well of normal growth media in an Agilent XF96 V3 PS Cell Culture microplate (Agilent). To achieve an even distribution of cells within wells, plates were incubated on the bench top at room temperature for 1 hr before incubating at 37°C, 5% $CO_2$ overnight. To hydrate the XF96 FluxPak (Agilent), 200 µL/well of sterile water was added and the entire cartridge was incubated at 37°C, $CO_2$-free incubator overnight. The following day, 1 hr prior to running the assay, 60 µL of media was removed, and the cells were washed twice with 200 µL/well of assay medium (XF DMEM Base Medium, pH 7.4 containing 25 mM glucose and 4 mM glutamine; Agilent). After washing, 160 µL/well of assay medium was added to the cell culture plate for a final volume of 180 µL/well. Cells were then incubated at 37°C, in a $CO_2$-free incubator until analysis. In parallel, 1 hr prior to the assay, water from the FluxPak hydration was exchanged for 200 µL/well of XF Calibrant 670 (Agilent), and the cartridge was returned to 37°C, $CO_2$-free incubator until analysis. Rotenone/Antimycin (50 µM, Agilent) and 2DG (500 mM, Agilent) were re-constituted in assay medium to make the indicated stock concentrations. 20 µL of rotenone/antimycin was loaded into Port A for each well of the FluxPak and 22 µL of 2DG into Port B, for a final concentration of 0.5 µM and 50 mM, respectively. The Glycolytic Rate Assay was conducted on an XF96 Extracellular Flux Analyzer (Agilent) and PER was calculated using Wave 2.6 software (Agilent). Following the assay, PER was normalized to cell number with the CyQUANT NF Cell Proliferation Assay (Invitrogen) according to manufacturer's instructions.

## Protein lysates

Cell lines cultured in 6 well plates in vitro were washed with ice-cold PBS on ice and incubated in 250 µL of RIPA buffer (Sigma) containing protease (Roche) and phosphatase (Sigma) inhibitors on ice for 10 min. Next, cells were scraped with a pipet tip, and the resulting lysate was transferred to a 1.5 mL tube also on ice. The lysate was centrifuged at 15,000 rpm for 10 min at 4°C. After, the supernatant was transferred to a fresh 1.5 mL tube and stored at –80°C.

In vivo, tumor tissue was placed in a 1.5 mL tube containing a metal ball and 300 µL RIPA buffer with protease and phosphatase inhibitors. The tissue was homogenized using a tissue lyser machine. Then, the resulting lysate was centrifuged at 15,000 rpm for 10 min at 4°C. After, the supernatant was transferred to a fresh 1.5 mL tube and stored at –80°C.

## Western blotting

Protein levels were determined using a BCA assay (Thermo Fisher), according to manufacturer's instructions. Following quantification, the necessary volume of lysate containing 30 µg of protein was added to a mixture of loading dye (Invitrogen) and reducing agent (Invitrogen) and incubated at 90°C for 5 min. Next, the lysate was separated on a 4–12% Bis-Tris gradient gel (Invitrogen) along with a protein ladder (Invitrogen) at 150 V until the dye reached the bottom of the gel (about 90 min). Then, the protein was transferred to a methanol-activated PVDF membrane (Millipore) at 25 V for 1 hr. After that, the membrane was blocked in 5% blocking reagent (Biorad) dissolved in TBS-T on a plate rocked for >1 hr. The membrane was then incubated overnight at 4°C rocking in the indicated primary antibody diluted in blocking buffer. The next day, the primary antibody was removed, and the membrane was washed three times in TBS-T rocking for 5 min. Then, the membrane was incubated for 1 hr rocking at room temperature in the appropriate secondary antibody diluted in TBS-T. Finally, the membrane was washed as before and incubated in Clarity ECL reagent (Biorad) according to manufacturer's instructions before imaging on a Biorad Chemidoc. The following primary antibodies were used in this study: GOT2 (Atlas, HPA018139), GOT1 (Abcam, ab171939), GLUD1 (Abcam, ab166618), IDH1 (Cell Signaling, 3997 S), MCT1 (Abcam, ab85021), MCT4 (Sigma, AB3316P), anti-Flag (Sigma, F3165), VINCULIN (Cell Signaling, 13,901 S), LDHA (Cell Signaling, 3582), LDHB (Abcam, ab53292), and the anti-rabbit-HRP secondary antibody (Cell Signaling, 7074 S).

## Isolating polar metabolites

For intracellular metabolome analyses, cells were seeded at 10,000 cells in 2 mL of growth media per well of a 6 well plate and incubated overnight. The next day, the growth media was removed, and cells were incubated in media containing the indicated compounds for 6 days, with the media being changed every 3 days. On day 6, the media was removed, and the cells were fixed and metabolites extracted into 1 mL/well of ice-cold 80% methanol on dry ice for 10 min. Following the incubation, the wells were scraped with a pipet tip and transferred to a 1.5 mL tube on dry ice.

To analyze extracellular metabolomes, 0.8 mL of ice-cold 100% methanol was added to 0.2 mL of media, mixed well, and incubated on dry ice for 10 min.

Mouse serum and TIF were isolated and analyzed as described previously (*Sullivan et al., 2019*).

The tubes were then centrifuged at 15,000 rpm for 10 min at 4°C to pellet insoluble material, and the resulting metabolite supernatant was transferred to a fresh 1.5 mL tube. The metabolites were then dried on a SpeedVac until the methanol/water had evaporated, and the resulting pellet was re-suspended in a 50:50 mixture of methanol and water.

## Snapshot metabolomics

Samples were run on an Agilent 1290 Infinity II LC –6470 Triple Quadrupole (QqQ) tandem mass spectrometer (MS/MS) system with the following parameters: Agilent Technologies Triple Quad 6470 LC-MS/MS system consists of the 1290 Infinity II LC Flexible Pump (Quaternary Pump), the 1290 Infinity II Multisampler, the 1290 Infinity II Multicolumn Thermostat with 6 port valves, and the 6470 triple quad mass spectrometer. Agilent Masshunter Workstation Software LC/MS Data Acquisition for 6400 Series Triple Quadrupole MS with Version B.08.02 is used for compound optimization, calibration, and data acquisition.

Solvent A is 97% water and 3% methanol 15 mM acetic acid and 10 mM tributylamine at pH of 5. Solvent C is 15 mM acetic acid and 10 mM tributylamine in methanol. Washing Solvent D is acetonitrile. LC system seal washing solvent 90% water and 10% isopropanol, needle wash solvent 75% methanol, 25% water. GC-grade Tributylamine 99% (ACROS ORGANICS), LC/MS grade acetic acid Optima (Fisher Chemical), InfinityLab Deactivator additive, ESI–L Low concentration Tuning mix

(Agilent Technologies), LC-MS grade solvents of water, and acetonitrile, methanol (Millipore), isopropanol (Fisher Chemical).

An Agilent ZORBAX RRHD Extend-C18, 2.1 × 150 mm and a 1.8 µm and ZORBAX Extend Fast Guards for ultra high performance liquid chromatography (UHPLC) are used in the separation. LC gradient profile is: at 0.25 mL/min, 0–2.5 min, 100% A; 7.5 min, 80% A and 20% C; 13 min 55% A and 45% C; 20 min, 1% A and 99% C; 24 min, 1% A and 99% C; 24.05 min, 1% A and 99% D; 27 min, 1% A and 99% D; at 0.8 mL/min, 27.5–31.35 min, 1% A and 99% D; at 0.6 mL/min, 31.50 min, 1% A and 99% D; at 0.4 mL/min, 32.25–39.9 min, 100% A; and at 0.25 mL/min, 40 min, 100% A. Column temperature is kept at 35°C, samples are at 4°C, injection volume is 2 µL.

The 6470 Triple Quad MS is calibrated with the Agilent ESI-L Low concentration Tuning mix. Source parameters: gas temperature 150°C, gas flow 10 L/min, nebulizer 45 psi, sheath gas temperature 325°C, sheath gas flow 12 L/min, capillary –2000 V, and delta EMV –200 V. Dynamic multiple reaction monitoring (dMRM) scan type is used with 0.07 min peak width, acquisition time is 24 min. dMRM transitions and other parameters for each compounds are list in a separate sheets. Delta retention time of plus and minus 1 min, fragmentor of 40 eV, and cell accelerator of 5 eV are incorporated in the method.

The MassHunter Metabolomics Dynamic MRM Database and method was used for target identification. The QqQ data were pre-processed with Agilent MassHunter Workstation QqQ Quantitative Analysis Software (B0700). Each metabolite abundance level in each sample was divided by the median of all abundance levels across all samples for proper comparisons, statistical analyses, and visualizations among metabolites. Metabolites with values >1 are higher in the experimental conditions, and metabolites with values <1 are lower in the experimental condition. The statistical significance test was done by a two-tailed t-test with a significance threshold level of 0.05.

Heatmaps were generated and data clustered using Morpheus Matrix Visualization and analysis tool (https://software.broadinstitute.org/morpheus).

Pathway analyses were conducted using MetaboAnalyst (https://www.metaboanalyst.ca).

## U13C-glucose, U13C-pyruvate isotope tracing

For glucose tracing, CAFs were seeded in 6 well plates at $2 \times 10^5$ cells/well and incubated for 72 hr in growth media containing U13C-glucose (Cambridge Isotope Laboratories).

For pyruvate tracing, PaTu-8902 and MIAPaCa-2 iDox-shGOT2.1 cells were cultured in media containing 1 mM unlabeled glucose and 1 mM U13C-pyruvate (Cambridge Isotope Laboratories) for 16 hr.

Polar metabolites were extracted from the media and cells according to the method described above. Isotope tracing experiments utilized the same chromatography as described in the Snapshot Metabolomics section and were conducted on two instruments with the following parameters:

Agilent Technologies Q-TOF 6530 LC/MS system consists of a 1290 Infinity II LC Flexible Pump (Quaternary Pump), 1290 Infinity II Multisampler, 1290 Infinity II Multicolumn Thermostat with 6 port valves, and a 6530 Q-TOF mass spectrometer with a dual-Assisted Jet Stream (AJI) ESI source. Agilent MassHunter Workstation Software LC/MS Data Acquisition for 6200 series TOF/6500 series Q-TOF Version B.09.00 Build 9.0.9044 .a SP1 is used for calibration and data acquisition.

Agilent 6530 Q-TOF MS is calibrated with ESI-L Low Concentration Tuning mix. Source parameters: gas temp 250°C, gas flow 13 L/min, nebulizer 35 psi, sheath gas temp 325° C, sheath gas flow 12 L/min, Vcap 3500 V, Nozzle Voltage (V) 1500, Fragmentor 140, Skimmer1 65, and OctopoleRFPeak 750. The MS acquisition mode is set in MS1 with mass range between 50 and 1200 da with collision energy of zero. The scan rate (spectra/s) is set at 1 Hz. The LC-MS acquisition time is 18 min and the total run time is 30 min. Reference masses are enabled with reference masses in negative mode of 112.9856 and 1033.9881 da.

Agilent Technologies 6545B Accurate-Mass Quadrupole Time of Flight (MS Q-TOF) LC/MS coupled with an Agilent 1290 Infinity II UHPLC. Agilent Masshunter Workstation Software LC/MS Data Acquisition for 6500 Series QTOF MS with Version 09.00, Build 9.0.9044.0 was used for tuning, calibration, and data acquisition.

In negative mode, the UHPLC was configured with 1290 Infinity II LC Flexible Pump (Quaternary Pump), 1290 Infinity II Multisampler, 1290 Infinity II Multicolumn Thermostat with 6 port valves. In

negative scan mode, the Agilent G6545B Q-TOF MS with Dual AJD ESI Sources in centroid mode was configured with following parameters: acquisition range: 50–1200 da at scan rate of 1 spectra/s, gas temp 250°C, gas flow 13 L/min, nebulizer at 40 psi, sheath gas heater 325°C, sheath gas flow 12 L/min, capillary 3500 V, nozzle voltage 1000 V, Fragmentor 130 V, Skimmer1 60 V, Octopole RFPeak 750 V, Collision 0 V, auto recalibration limit of detection 150 ppm with minute height of 1000 counts, and reference ions of two at 59.0139 and 980.0164 da.

Data processing was performed in Agilent MassHunter Workstation Profinder 10.0 Build 10.0.10062.0. Isotopologue distributions were derived from a compound standard library built in Agilent MassHunter PCDL (Personal Compound and Database Library) v7.0.

## Xenograft studies

Animal experiments were conducted in accordance with the Office of Laboratory Animal Welfare and approved by the Institutional Animal Care and Use Committees of the University of Michigan. The NSG mice (Jackson Laboratory) 6–10 weeks old of both sexes were maintained in the facilities of the Unit for Laboratory Animal Medicine under specific pathogen-free conditions.

Cells expressing doxycycline-inducible shNT or doxycycline-inducible shGOT2 were injected subcutaneously into both the left and right flanks of male and female NSG mice, with 1–4 × 10⁶ cells in a mixture of 50 µL media and 50 µL Matrigel (Corning) per injection. Tumors were established for 7 days before mice were fed either normal chow or chow containing doxycycline (BioServ). Tumors were measured with calipers two times per week, and mice were euthanized once the tumors reached a diameter of 2 cm³. Subcutaneous tumor volume (V) was calculated as V = 1/2 (length × width; *Vander Heiden and DeBerardinis, 2017*). At endpoint, the tumors were removed, and fragments were either snap frozen in liquid nitrogen and stored at –80°C or fixed in ZFix solution (Anatech) for histology.

Cells expressing doxycycline-inducible shGOT2 and luciferase were injected into the pancreas tail of NSG mice, with 200,000 cells in a mixture of 50 µL media and 50 µL Matrigel (Corning) per injection. Tumors were established for 7 days before mice were fed either normal chow or chow containing doxycycline (BioServ). Tumor progression was monitored by weekly intraperitoneal injections of luciferin (Promega) and bioluminescence imaging (BLI) on an IVIS SpectrumCT (Perkin Elmer). BLI was analyzed with Living Image software (PerkinElmer) At endpoint, the tumors were removed, and fragments were either snap frozen in liquid nitrogen and stored at –80°C or fixed in ZFix (Anatech) solution for histology.

AZD3965 or FX11 was dissolved in DMSO and stored at –80°C in aliquots. Each day, one aliquot was thaw and mixed with a 0.5% Hypromellose (Sigma), 0.2% tween 80 (Sigma) solution such that the final DMSO concentration was 5%. Vehicle or AZD3965 was administered at 100 mg/kg by daily oral gavage. FX11 was administered at 2 mg/kg by daily intraperitoneal injection.

## Histology

Tissues were processed using a Leica ASP300S tissue processor (Leica Microsystems). Paraffin-embedded tissues were sectioned at 4 µm and stained for specific target proteins using the Discovery Ultra XT autostainer (Ventana Medical Systems), with listed antibodies, and counterstained with Mayer's hematoxylin (Sigma). H&E staining was performed using Mayer's hematoxylin solution and Eosin Y (Thermo Fisher). IHC slides were then scanned on a Pannoramic SCAN scanner (Perkin Elmer). Scanned images were quantified using algorithms provided from Halo software version 2.0 (Indica Labs). The following antibodies were used for IHC: Ki67 (1:1000; Abcam, ab15580), αSMA (1:20,000; Abcam, ab5694), and Got2 (1:500; Atlas, HPA018139).

## KC-Got2^(f/f) model

Mice containing loxP sites flanking exon 2 of the Got2 gene were generated by Ozgene. These mice were crossed to the LSL-Kras^(G12D);Ptf1a-Cre model (*Hingorani et al., 2003*). Tails from 3-week-old mice were collected at weaning and submitted to Transnetyx for genotyping. The following primers were used: Kras^(G12D) (Fw-GGCCTGCTGAAAATGACTGAGTATA, Rev-CTGTATCGTCAAGGCGCTCTT); Got2 WT (Fw- GCAGATTAAAACCACAAGGCCTGTA, Rev-ATGTTAAAATTGTCATCCCCTTGTGC); Got2

floxed (Fw-GCAGATTAAAACCACAAGGCCTGTA, Rev-AGAGAATAGGAACTTCGGAATAGGAACT); Cre (Fw-TTAATCCATATTGGCAGAACGAAAACG, Rev-CAGGCTAAGTGCCTTCTCTACA).

## Statistics

Statistics were performed using Graph Pad Prism 8. Groups of 2 were analyzed with two-tailed students t test, groups greater than 2 were compared using one-way ANOVA analysis with Tukey post hoc test or two-way ANOVA with Dunnett's correction for multiple independent variables. All error bars represent mean with SD, all group numbers and explanation of significant values are presented within the figure legends. Experiments were repeated at least twice to verify results.

## Acknowledgements

This work was funded by T32AI007413, F31CA24745701, and 1F99CA264414-01 (SAK); CA148828 and CA245546 (YMS); a Pancreatic Cancer Action Network/AACR Pathway to Leadership award (13-70-25-LYSS), Junior Scholar Award from The V Foundation for Cancer Research (V2016-009), Kimmel Scholar Award from the Sidney Kimmel Foundation for Cancer Research (SKF-16–005), a 2017 AACR NextGen Grant for Transformative Cancer Research (17-20-01-LYSS), and NIH grants R37CA237421, R01CA248160, R01CA244931 (CAL). Additional funding sources include F31CA254079 (JAJ); T32-DK094775 and T32-CA009676 (BSN); R50 CA232985 (YZ), T32-GM11390 and F31-CA247076 (SBK); T32-CA009676, the American Cancer Society Postdoctoral Award PF-19-096-01, and the Michigan Institute for Clinical and Healthy Research (MICHR) Postdoctoral Translational Scholar Program fellowship award (NGS); the Michigan Postdoctoral Pioneer Program (ZCN); K99CA241357 and P30DK034933 (CJH); R01GM101171 and CA253986 (DBL); and Cancer Center support grant (P30 CA046592). Metabolomics studies performed at the University of Michigan were supported by NIH grant DK097153, the Charles Woodson Research Fund, and the UM Pediatric Brain Tumor Initiative. Research reported in this publication was supported by the National Cancer Institutes of Health under Award Number P30CA046592 by the use of the following Cancer Center Shared Resource(s): Flow Cytometry Core, Tissue and Molecular Pathology Core, Center for Molecular Imaging (P30CA046592). The University of Michigan Center for Gastrointestinal Research Core Director, Michael Mattea, provided assistance in tissue processing, sectioning, and staining.

## Additional information

### Funding

| Funder | Grant reference number | Author |
| --- | --- | --- |
| National Institute of Allergy and Infectious Diseases | T32AI007413 | Samuel A Kerk |
| National Cancer Institute | F31CA24745701 | Samuel A Kerk |
| National Cancer Institute | 1F99CA264414-01 | Samuel A Kerk |
| National Cancer Institute | CA148828 | Yatrik M Shah |
| National Cancer Institute | CA245546 | Yatrik M Shah |
| Pancreatic Cancer Action Network | 13-70-25-LYSS | Costas A Lyssiotis |
| V Foundation for Cancer Research | V2016-009 | Costas A Lyssiotis |
| Sidney Kimmel Foundation | SKF-16-005 | Costas A Lyssiotis |
| American Association for Cancer Research | 17-20-01-LYSS | Costas A Lyssiotis |
| National Cancer Institute | R37CA237421 | Costas A Lyssiotis |

| Funder | Grant reference number | Author |
| --- | --- | --- |
| National Cancer Institute | R01CA248160 | Costas A Lyssiotis |
| National Cancer Institute | R01CA244931 | Costas A Lyssiotis |
| National Cancer Institute | F31CA254079 | Jennifer A Jiménez |
| National Institute of Diabetes and Digestive and Kidney Diseases | T32-DK094775 | Barbara S Nelson |
| National Cancer Institute | T32-CA009676 | Barbara S Nelson Nina G Steele |
| National Cancer Institute | R50 CA232985 | Yaqing Zhang |
| National Institute of General Medical Sciences | T32-GM11390 | Samantha B Kemp |
| National Cancer Institute | F31-CA247076 | Samantha B Kemp |
| American Cancer Society | PF-19-096-01 | Nina G Steele |
| National Cancer Institute | K99CA241357 | Christopher J Halbrook |
| National Institute of Diabetes and Digestive and Kidney Diseases | P30DK034933 | Christopher J Halbrook |
| National Institute of General Medical Sciences | R01GM101171 | David B Lombard |
| National Cancer Institute | CA253986 | David B Lombard |
| National Cancer Center | P30 CA046592 | Costas A Lyssiotis |

The funders had no role in study design, data collection and interpretation, or the decision to submit the work for publication.

### Author contributions

Samuel A Kerk, Conceptualization, Formal analysis, Investigation, Methodology, Writing – original draft, Writing – review and editing; Lin Lin, Amy L Myers, Anthony Andren, Peter Sajjakulnukit, Li Zhang, Yaqing Zhang, Jennifer A Jiménez, Barbara S Nelson, Brandon Chen, Anthony Robinson, Galloway Thurston, Samantha B Kemp, Nina G Steele, Megan T Hoffman, Hui-Ju Wen, Daniel Long, Zeribe C Nwosu, Christopher J Halbrook, Data curation, Formal analysis; Damien J Sutton, Formal analysis, Investigation; Sarah E Ackenhusen, Johanna Ramos, Xiaohua Gao, Data curation; Stefanie Galban, Haoqiang Ying, Resources; David B Lombard, Marina Pasca di Magliano, Resources, Project administration; David R Piwnica-Worms, Resources, Writing – review and editing; Howard C Crawford, Resources, Data curation, Formal analysis, Project administration; Yatrik M Shah, Conceptualization, Resources, Writing – original draft, Project administration, Writing – review and editing; Costas A Lyssiotis, Conceptualization, Resources, Formal analysis, Supervision, Funding acquisition, Investigation, Methodology, Writing – original draft, Project administration, Writing – review and editing

### Author ORCIDs

Samuel A Kerk ![ORCID] http://orcid.org/0000-0001-9786-2245
Marina Pasca di Magliano ![ORCID] http://orcid.org/0000-0001-9632-9035
Costas A Lyssiotis ![ORCID] http://orcid.org/0000-0001-9309-6141

### Ethics

Animal experiments were conducted in accordance with the Office of Laboratory Animal Welfare and approved by the Institutional Animal Care and Use Committees of the University of Michigan. ULAM: PRO00008877.

### Decision letter and Author response

Decision letter https://doi.org/10.7554/eLife.73245.sa1
Author response https://doi.org/10.7554/eLife.73245.sa2

## Additional files

### Supplementary files

• Supplementary file 1. Metabolomics data from *Figure 1—figure supplement 2A*, *Figure 3—figure supplement 1D*, and *Figure 5—figure supplement 1E*.

• Transparent reporting form

### Data availability

All data generated or analyzed during this study are included in the manuscript and supporting source data. Raw data are provided for metabolomics experiments, separated by tabs, in Supplementary file 1.

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
