## [Editor Report]

This paper provides evidence that the environmental redox state shapes metabolic liabilities in pancreatic cancer cells. In vitro, pancreatic cancer cells rely on the malate-aspartate shuttle to regenerate oxidized electron acceptors in the cytosol that are required for cell proliferation; stromal cells within tumors secrete metabolites that serve as alternate electron acceptors, thereby reducing cancer cell reliance on the malate-aspartate shuttle.

---

## [Decision Letter]

**Decision letter after peer review:**

Thank you for submitting your article "GOT2 is Dispensable for Pancreatic Tumor Growth" for consideration by *eLife*. Your article has been reviewed by 3 peer reviewers, one of whom is a member of our Board of Reviewing Editors, and the evaluation has been overseen by Richard White as the Senior Editor. The reviewers have opted to remain anonymous.

Essential revisions:

1. The authors should restructure the manuscript. The major issue is that all experiments to probe the role of pyruvate import and reduction to lactate in vivo provide negative results. While all reviewers commended the authors for including the data, and agreed that other factors could contribute to these negative results, at present there is no reason to conclude that environmental pyruvate and/or changes in redox state enable tumors to grow in vivo without GOT2. If anything, MCT1 depletion seems to promote tumor growth, arguing against the favored hypothesis. Alternative explanations, such as a potentially reduced reliance on glutamine for TCA cycle anaplerosis in vivo, are not considered. Therefore, the central conclusion of the manuscript remains largely unsupported. Absent supportive evidence, the manuscript should be largely restructured to remove any claims as to the role of the TME supporting redox homeostasis in cells lacking GOT2 in vivo. The authors perhaps could refocus the manuscript to show how their data provide a striking example of how the metabolic dependencies identified in culture may not translate to the complex microenvironment of tumors, as suggested by Reviewer #2. All reviewers agreed that it is very important that the authors ensure it is clear in abstract and other places that it remains unresolved whether CAFs actually contribute to redox balance in the tumor, even though they hold this potential as shown in the cell culture experiments.

2. Overall, alternative explanations are not often tested. For example, the experiments showing that pyruvate and LbNOX rescue GOT2 deficiency are interesting and certainly support the notion that cytosolic redox is limiting for proliferation following GOT2 deficiency. However, an important control is assessing whether these interventions rescue any metabolic consequences of GOT2 loss (e.g. aspartate, glutamine anaplerosis, etc). This point is important as Extended Data 4d shows a nearly identical rescue between pyruvate and αKG+asp, aside from one replicate being higher. Similarly, the conditioned medium and pyruvate rescues are very clear, but the aKB and LbNOX are more partial and the LDH inhibitor effect is fairly modest. Is pyruvate doing something more in addition to impacting redox homeostasis?

3. Related, the authors should also test whether macropinocytosis, known to be important in PDAC tumors in vivo, can compensate for GOT2 loss. Perhaps macropinocytosis is impaired in culture due to lack of protein availability (or other environmental causes) rendering cells sensitive to GOT2 loss, but activation of macropinocytosis in vivo may cause GOT2 to be dispensable. Have the authors investigated if macropinocytosis and GOT2 metabolism are two parallel paths of nutrient sufficiency? For example, would additional BSA rescue GOT2-deficient cells in vitro? Linking the phenotypes of GOT2 deficient cells with the pancreatic cancer macropinocytosis literature would considerably increase the impact of this work.

4. What is the relative concentration of pyr/lac in conditioned medium in vitro, and in the serum/TME in vivo? Of note, authors mention that mouse plasma has ~250 μm pyruvate, but the same study (Sullivan et al. 2019) found that tumor interstitial fluid had quite a bit less pyruvate than plasma. It is possible that lactate is much more abundant than pyruvate, and so the relative degree to which the pyruvate/lactate exposure in CM can be net oxidizing remains untested. This becomes even more important given the finding that MCT1 depletion seems to enhance tumor growth. To address this point, the authors could measure pyruvate and lactate in CAF conditioned media and test if adding pyr/lac at these relevant levels is sufficient to rescue GOT2 loss.

5. To what extent does MCT1 KO block pyruvate uptake? Are there any changes in MCT4 expression upon MCT1 KO? If feasible, it would be nice to see whether a double MCT1/4 KO would prevent tumor growth in the context of GOT2 KD, but this latter experiment is not necessary for publication.

---

## [Author Response]

Essential revisions:1. The authors should restructure the manuscript. The major issue is that all experiments to probe the role of pyruvate import and reduction to lactate in vivo provide negative results. While all reviewers commended the authors for including the data, and agreed that other factors could contribute to these negative results, at present there is no reason to conclude that environmental pyruvate and/or changes in redox state enable tumors to grow in vivo without GOT2. If anything, MCT1 depletion seems to promote tumor growth, arguing against the favored hypothesis. Alternative explanations, such as a potentially reduced reliance on glutamine for TCA cycle anaplerosis in vivo, are not considered. Therefore, the central conclusion of the manuscript remains largely unsupported. Absent supportive evidence, the manuscript should be largely restructured to remove any claims as to the role of the TME supporting redox homeostasis in cells lacking GOT2 in vivo. The authors perhaps could refocus the manuscript to show how their data provide a striking example of how the metabolic dependencies identified in culture may not translate to the complex microenvironment of tumors, as suggested by Reviewer #2. All reviewers agreed that it is very important that the authors ensure it is clear in abstract and other places that it remains unresolved whether CAFs actually contribute to redox balance in the tumor, even though they hold this potential as shown in the cell culture experiments.

We agree with this important critique of the previous manuscript, and we appreciate the associated suggestions on how to reframe our work. Herein, we have restructured the manuscript to emphasize the redox mechanism involving GOT2 knockdown (KD) in vitro, and we have reframed our results to highlight differences between in vitro and in vivo metabolism. A section in the discussion on previous work investigating glutamine metabolism in vitro versus in vivo has also been added (lines 387-392). The language regarding the role of the TME in maintaining redox balance has been appropriately tempered, and, rather, we posit in the discussion that the proposed mechanism could be one of several potential mechanisms driving GOT2 independent growth in vivo. Further, we have made sure to clearly detail in the abstract, text accompanying associated experiments, as well as in the discussion (line 429-430), that the role of CAFs in redox balance in vivo remains to be investigated. Finally, we have rewritten the abstract to (i) focus on the in vitro work identifying how GOT2 KD induces NADH stress, a role we think has been previously overshadowed due to justifiable interest in aspartate metabolism, and (ii) illustrate the complexity of in vivo metabolism, especially for essential metabolic concepts like redox balance.

2. Overall, alternative explanations are not often tested. For example, the experiments showing that pyruvate and LbNOX rescue GOT2 deficiency are interesting and certainly support the notion that cytosolic redox is limiting for proliferation following GOT2 deficiency. However, an important control is assessing whether these interventions rescue any metabolic consequences of GOT2 loss (e.g. aspartate, glutamine anaplerosis, etc). This point is important as Extended Data 4d shows a nearly identical rescue between pyruvate and αKG+asp, aside from one replicate being higher. Similarly, the conditioned medium and pyruvate rescues are very clear, but the aKB and LbNOX are more partial and the LDH inhibitor effect is fairly modest. Is pyruvate doing something more in addition to impacting redox homeostasis?

This is absolutely an important point, and we appreciate the opportunity to include the necessary control data. Herein we provide new metabolomics data from GOT2KD + EV, LbNOX, or mLbNOX. These data illustrate that LbNOX, but not mLbNOX, resolves the NADH/NAD imbalance (Figure 2H, Figure 2—figure supplement 2D). Similarly, the secreted pyruvate/lactate ratio indicates specificity for the cytosolic pools (Figure 2—figure supplement 2E,F). In parallel, we observe that LbNOX, but not mLbNOX, rescues the metabolic defects observed with GOT2 KD. The stalled glycolytic signature is reversed, TCA cycle metabolite levels are corrected, and both Asp and αKG levels increase (Figure 2—figure supplement 2G).

To address the rescue discrepancy between the pyruvate and Asp+αKG rescues, we repeated the experiment using serum concentrations reported in mice (PMC6510537); i.e. pyruvate (250 µM), Asp (50 µM), and αKG (500 µM). For αKG, we employed the esterified dimethyl-αKG analog to increase the intracellular availability. Here, we again demonstrate that pyruvate rescued to a greater extent than Asp+αKG (Figure 5—figure supplement 2B). Considering these levels of Asp and αKG reported in mouse serum are significantly higher than those measured in CAF CM, we conclude that Asp+αKG are not the responsible components of CAF CM for rescue activity. That said, we do find that supraphysiological Asp (20 mM) and dimethyl-αKG (1 mM) rescue GOT2 KD growth (Figure 1F).

We also repeated the αKB (Figure 2F) and LbNOX GOT2 KD (Figure 2G) rescue experiments to show the extent to which these interventions promote proliferation compared to pyruvate.

Lastly, we repeated the FX11+GOT2 KD rescue experiments with pyruvate to demonstrate that FX11 strongly inhibits the pyruvate-mediated rescue of GOT2 KD in vitro (Figure 6H).

Collectively, these data continue to support our model that the primary in vitro mechanism by which pyruvate rescues GOT2 KD is via NADH turnover, through LDHA, which reverses reductive stress and allows cellular metabolism to resume. In further support of this model, our 13C3-Pyruvate tracing data presented in Figure 2I-K show that supplemented pyruvate is rapidly converted to lactate in GOT2 KD cells. Importantly, while blocking pyruvate entry to the mitochondria via MPC inhibition with UK5099 decreases pyruvate carbon incorporation into the TCA cycle (Figure 2—figure supplement 3B-D), this has no effect on the GOT2 KD rescue of pyruvate supplementation (Figure 2—figure supplement 3E). These data suggest that the activity of pyruvate is non-mitochondrial. We also show data to suggest that pyruvate conversion to lactate (Figure 2—figure supplement 3F) or alanine (Figure 2—figure supplement 3G) is not relevant for our mechanism, as these metabolites provide no rescue activity to GOT2 KD cells.

3. Related, the authors should also test whether macropinocytosis, known to be important in PDAC tumors in vivo, can compensate for GOT2 loss. Perhaps macropinocytosis is impaired in culture due to lack of protein availability (or other environmental causes) rendering cells sensitive to GOT2 loss, but activation of macropinocytosis in vivo may cause GOT2 to be dispensable. Have the authors investigated if macropinocytosis and GOT2 metabolism are two parallel paths of nutrient sufficiency? For example, would additional BSA rescue GOT2-deficient cells in vitro? Linking the phenotypes of GOT2 deficient cells with the pancreatic cancer macropinocytosis literature would considerably increase the impact of this work.

We recognize the importance of this suggestion. As we now discuss in lines 416-418, macropinocytosis could be an in vivo mechanism by which cells obtain Asp or pyruvate, though it should be noted that we observe significantly reduced Asp levels in vivo (Figure 3E). Indeed, we and others have demonstrated that macropinocytosis is highly active in PDA cell lines in vitro (PMC8730721, PMC3810415). In standard culture conditions, our PDA cells are maintained in 10% FBS, providing adequate protein for these purposes. Despite high basal macropinocytosis and adequate protein, we still see a strong GOT2 KD phenotype in vitro.

That said, we appreciate the spirit of suggestion, and to test this possibility, we cultured GOT2 KD cells in +/- 2% BSA, conditions that rescue glutamine deprivation in PDA cells via micropinocytosis (PMC3810415). BSA did not provide a growth rescue of GOT2 KD (Figure 6—figure supplement 3C).

We would like to reiterate that we are not ruling out the possibility that macropinocytosis could compensate for GOT2 loss in vivo. Nevertheless, we have included these BSA rescue data and the indicated section in the discussion because macropinocytosis is clearly an important mechanism for nutrient acquisition in PDA.

4. What is the relative concentration of pyr/lac in conditioned medium in vitro, and in the serum/TME in vivo? Of note, authors mention that mouse plasma has ~250 μm pyruvate, but the same study (Sullivan et al. 2019) found that tumor interstitial fluid had quite a bit less pyruvate than plasma. It is possible that lactate is much more abundant than pyruvate, and so the relative degree to which the pyruvate/lactate exposure in CM can be net oxidizing remains untested. This becomes even more important given the finding that MCT1 depletion seems to enhance tumor growth. To address this point, the authors could measure pyruvate and lactate in CAF conditioned media and test if adding pyr/lac at these relevant levels is sufficient to rescue GOT2 loss.

We appreciate this excellent suggestion. To test the rescue activity of pyruvate and pyruvate to lactate ratios at context relevant concentrations, we first determined the absolute concentration of pyruvate and lactate in CAF CM (Figure 6A) and mouse serum and TIF (Figure 6B). We did not observe a significant difference in pyruvate levels between TIF and serum in this model, though we did detect a significant increase in lactate, as would be expected in a tumor. Next, we supplemented growth media with the measured concentration of pyruvate and lactate in CAF CM, serum, and TIF to artificially recreate the pyruvate/lactate ratios of these environments. Each of these levels and ratios promoted growth of GOT2 KD cells in vitro, effectively in a manner akin to pyruvate alone (Figure 6C).

5. To what extent does MCT1 KO block pyruvate uptake?

To determine the impact of MCT1 KO on pyruvate uptake, we incubated MCT1-expressing (EV) or MCT1 KO (sg1, sg2) cells with 13C3-Pyruvate and measured intracellular pyruvate and lactate by LC-MS/MS over a 5 minute window, a well appreciated time frame for pyruvate uptake/metabolism by the hyperpolarization community (PMID: 29769199, PMID: 32466260). First, we observed a decrease in labelled pyruvate in MCT1 KO cells (Figure 6G). While we washed the cells of media (and label) prior to collection, we put forth that the modest difference in labeled pyruvate results from media contamination. Indeed, as is far more evident, we observed substantially less labelled lactate in MCT1 KO cells (Figure 6G), a reflection of both pyruvate uptake and NADH turnover. These data confirm that MCT1 KO impairs pyruvate metabolism in PDA cell lines.

Are there any changes in MCT4 expression upon MCT1 KO?

While GOT2 KD MCT1 KO flank xenografts do express MCT4, we did not observe a further compensatory increase in MCT4 (Figure 6—figure supplement 2F). Similarly, we also do not observe an increase in MCT4 expression in MCT1 KO cell lines in vitro (Figure 6—figure supplement 2G).

If feasible, it would be nice to see whether a double MCT1/4 KO would prevent tumor growth in the context of GOT2 KD, but this latter experiment is not necessary for publication.

We agree with the referee that this is an excellent experiment. While we do not test dual inhibition of MCT1 and MCT4 in GOT2 KD flank xenografts in this study, we do discuss this mechanism in lines 392-397.